# Incomplete influenza A virus genomes occur frequently but are readily complemented during localized viral spread

Nathan T. Jacobs[1], Nina O. Onuoha[1], Alice Antia[1], John Steel[1,4], Rustom Antia[2] & Anice C. Lowen [1,3]

Segmentation of viral genomes into multiple RNAs creates the potential for replication of incomplete viral genomes (IVGs). Here we use a single-cell approach to quantify influenza A virus IVGs and examine their fitness implications. We find that each segment of influenza A/Panama/2007/99 (H3N2) virus has a 58% probability of being replicated in a cell infected with a single virion. Theoretical methods predict that IVGs carry high costs in a well-mixed system, as 3.6 virions are required for replication of a full genome. Spatial structure is predicted to mitigate these costs, however, and experimental manipulations of spatial structure indicate that local spread facilitates complementation. A virus entirely dependent on co-infection was used to assess relevance of IVGs in vivo. This virus grows robustly in guinea pigs, but is less infectious and does not transmit. Thus, co-infection allows IVGs to contribute to within-host spread, but complete genomes may be critical for transmission.

---

[1] Department of Microbiology and Immunology, Emory University School of Medicine, Atlanta, GA, USA. [2] Department of Biology, Emory University, Atlanta, GA, USA. [3] Emory-UGA Center of Excellence for Influenza Research and Surveillance, Emory University School of Medicine, Atlanta, GA, USA. [4] Present address: Influenza Division, Centers for Disease Control and Prevention, Atlanta, GA, USA. Correspondence and requests for materials should be addressed to A.C.L. (email: anice.lowen@emory.edu)

Pathogen evolution poses a continued threat to public health by reducing the effectiveness of antimicrobial drugs and adaptive immunity. In the case of the influenza A virus (IAV), this evolution results in seasonal outbreaks as new viruses emerge to which preexisting immunity is weak. Each year requires a new vaccine as a consequence, and keeping pace with IAV evolution is challenging: unexpected emergence of new strains could render the vaccine obsolete before the flu season starts. IAV populations evolve rapidly in part because their mutation rates are high, on the order of $10^{-4}$ substitutions per nucleotide per genome copied[1]. The segmentation of the viral genome gives a second source of genetic diversity. The IAV genome is composed of eight single-stranded RNA segments, and so cells coinfected with two different IAV virions can produce chimeric progeny with a mix of segments from these two viruses. This process, termed reassortment, carries costs and benefits analogous to those of sexual reproduction in eukaryotes[2]. Reassortment can combine beneficial mutations from different backgrounds to alleviate clonal interference[3], and purge deleterious mutations to mitigate the effects of Muller's ratchet[4,5]. This combinatorial shuffling of mutations may accelerate adaptation to new environments such as a novel host[6]. Conversely, free mixing of genes through reassortment may also reduce viral fitness by separating beneficial segment pairings, as sexual reproduction carries this cost in eukaryotes[7]. Previous work has shown that reassortment occurs readily between closely related variants[8], but is limited between divergent lineages due to molecular barriers[9,10] or reduced fitness of progeny[11,12]. Nevertheless, the contribution of reassortment to emergence of novel epidemic and pandemic IAVs has been documented repeatedly[13–17]. Factors that affect the frequency of coinfection and consequently reassortment are therefore likely to play an important role in viral evolution.

While the ability of a virus particle to enter a cell depends only on the proteins that line the virion surface, subsequent production of viral progeny requires successful expression and replication of the genome. A virion that does not contain, or fails to deliver, a complete genome could therefore infect a cell but fail to produce progeny. IAV particles outnumber plaque-forming units (PFUs) by ~10–100-fold[18], meaning that only a minority of particles establish productive infection at limiting dilution. Recent data suggest that IAV infection is not a binary state, however. Efforts to detect viral proteins and mRNAs at the single-cell level have revealed significant heterogeneity in viral gene expression[19–22]. These data furthermore suggest that a subset of gene segments is often missing entirely from cells infected at low multiplicity of infection (MOI). Thus, many non-plaque-forming particles appear to be semi-infectious, giving rise to incomplete viral genomes (IVGs) within the infected cell[23].

Replication and expression of only a subset of the genome may be explained by two potential mechanisms: either the majority of particles lack one or more genome segments, or segments are readily lost in the process of infection before they can be replicated. Published data suggest that most particles contain full genomes: electron microscopy revealed eight distinct RNA segments in most virions[24], and FISH-based detection of viral RNAs indicated that a virion typically contains one copy of each segment[25]. Loss of segments following delivery of a viral genome to the target cell therefore seems likely to be an important mechanism. Inefficiencies inherent in the processes of cytoplasmic trafficking, nuclear import, and replication of incoming viral RNAs during the earliest stages of infection would all lead to loss of segments. Very likely, multiple mechanisms contribute to give rise to incomplete IAV genomes.

The frequent occurrence of incomplete IAV genomes suggests that complementation by cellular coinfection is an underappreciated aspect of the viral life cycle. The observation of appreciable levels of reassortment following coinfection at low MOIs suggested IVG reactivation through complementation occurs commonly during IAV infection[26], but the extent to which replication depends on complementation remains unclear. Similarly, the existence of IVGs in vivo has been demonstrated[27], but their importance to the dynamics of infection within hosts is untested.

Here, we investigate the biological implications of incomplete IAV genomes and the emergent need for cooperation at the cellular level. We first developed a single-cell sorting assay to measure the probability of each segment being delivered by an individual virion for influenza A/Panama/2007/99 (H3N2) [Pan/99] virus. Our data estimate that individual virus particles lead to successful replication of all eight gene segments only 1.22% of the time. When considering a well-mixed system in which virus particles are distributed randomly over cells, the potential fitness costs of incomplete genomes are high. In contrast, a model of viral spread that incorporates local dispersal of virions to nearby cells predicts that the spatial structure of virus growth mitigates costs of genome incompleteness. Testing of this model confirmed that infections initiated with randomly distributed inocula contain more IVGs than those generated by secondary spread from low MOIs, in which the spatial structure is inherent. To determine the potential for complementation to occur in vivo, we generated a mutant virus that required cellular coinfection for replication. We find that this virus is able to grow within guinea pigs, but unable to transmit to cagemates. These results suggest that the abundance of incomplete genomes and the potential for complementation are important factors in the replication and transmission of IAV.

## Results

**Measurement of $P_P$.** To evaluate the implications of genome incompleteness for IAV fitness and reassortment, we sought to quantify the probability of successful replication for each of the eight genome segments within single cells infected with single virions. To ensure accurate detection of IVGs, we devised a system that would allow their replication to high copy number. We applied our approach to the seasonal isolate, influenza A/Panama/2007/99 (H3N2) virus. In this assay, MDCK cells are inoculated with a virus of interest, referred to as "Pan/99-WT" or "WT", and a genetically tagged helper virus ("Pan/99-Helper" or "Helper"). This Helper virus differs from the WT strain only by silent mutations on each segment that provide distinct primer-binding sites. For example, qPCR primers targeting WT PB2 will not anneal to cDNA of Helper PB2, and vice versa. By co-inoculating cells with a low MOI of WT virus and a high MOI of Helper virus, we ensure that each cell is productively infected, but is unlikely to receive more than one WT virion. Following infection, one cell per well is sorted into a 96-well plate containing MDCK cell monolayers. The initially infected cell produces progeny which then infect neighboring cells, effectively amplifying the vRNA segments present in the first cell. The presence or absence of WT segments in each well can then be measured by segment-specific RT-qPCR. As detailed in the Methods section, the frequencies of Helper virus infection, WT virus infection, and each distinct WT segment were used to estimate the probability that a cell infected with a single WT virion would contain a given segment. The calculation used takes into account the known probability of multiple infection. We termed the resultant parameter "Probability Present", and refer to it hereafter as $P_P$. Segment-specific values are referred to as $P_{P,i}$, where $i = 1$–8, while $P_P$ refers to the average $P_P$ value across all segments, which is calculated as the geometric mean of eight

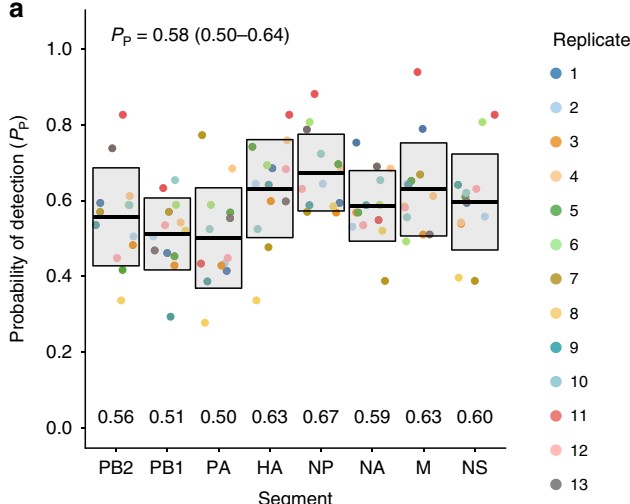

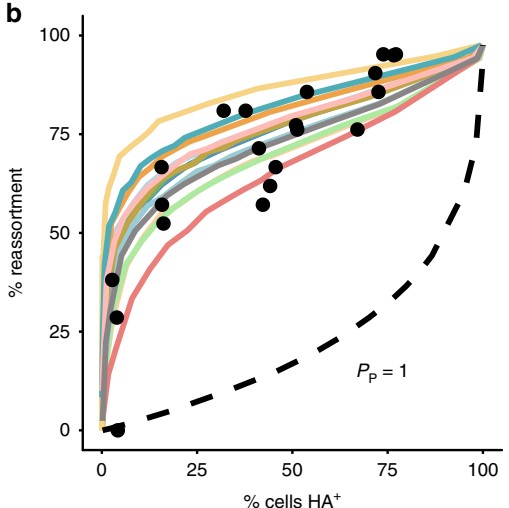

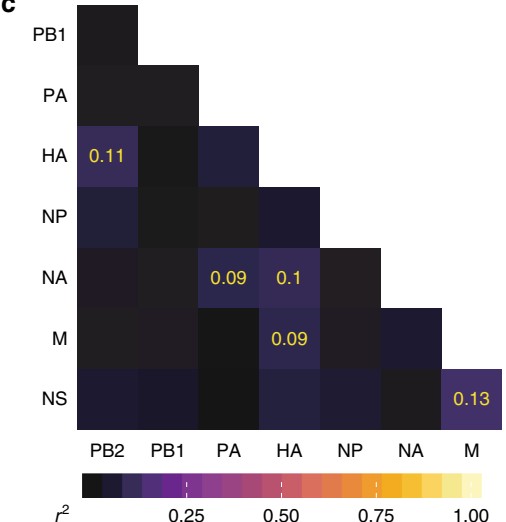

**Fig. 1** Incomplete genomes are common in Pan/99 virus infection.
**a** Segment-specific $P_{P,i}$ values were measured by a single-cell sorting assay. Each set of colored points corresponds to eight $P_{P,i}$ values measured in a single experimental replicate, with 13 independent replicates performed. Horizontal bars indicate the mean (written above each segment's name), and shading shows the mean ± SD ($N = 13$ independent experiments). **b** Using each replicate's $P_{P,i}$ values as input parameters, the computational model from Fonville et al. was used to predict the frequency of reassortment across multiple levels of infection[26]. Black circles represent the experimental data from Fonville et al. and show levels of reassortment observed following single-cycle coinfection of MDCK cells with Pan/99-WT and a Pan/99 variant viruses. Colored lines show the theoretical predictions made by the model, with colors corresponding to the legend shown in panel **a**. **c** Pairwise correlations between segments ($r^2$) are shown as color intensities represented by a color gradient (below). $r^2$ values are shown in yellow for significant associations ($Nr^2$ (where $N$ is the sum of $p$(1 virion) values, $N = 186$) follows a $\chi^2$ distribution with three degrees of freedom, $p < 0.05$ ($\chi^2$ test) after Bonferroni correction for multiple comparisons). Source data are provided as a Source Data file

When used to parameterize a model that estimates the frequency of reassortment[26], these $P_{P,i}$ values generated predictions that align closely with the experimental data (Fig. 1b). This match between observed and predicted reassortment is important because (i) it offers a validation of the measured $P_{P,i}$ values, and (ii) it indicates that IVGs fully account for the levels of reassortment observed, which are much higher than predicted for viruses with only complete genomes[26].

Interactions between viral ribonucleoprotein (vRNP) segments play an important role in the assembly of new virions[10,28–31]. To determine whether similar interactions exist that could mediate the co-delivery of segments to the cell, patterns of segment co-occurrence were analyzed. In this analysis, it was important to account for the probability of multiple infection in our single-cell assay. As shown in Supplementary Fig. 1, cells containing more segments were likely to have been infected with multiple virions, and so we applied a weighting factor to ensure that results relied more strongly on data from cells with fewer WT segments. Namely, we determined the probability that a given cell acquired its segment constellation by infection with a single virion, and weighted data according to this probability to calculate the pairwise correlation between segments. While some significant interactions were observed, they were relatively weak, with $r^2$ values below 0.15 (Fig. 1c). Thus, our data suggest that associations among specific vRNPs do not play a major role during the establishment of infection within a cell.

Given the independence of vRNP delivery and the similarity between $P_{P,i}$ values, we calculated an average $P_P$ value for use in subsequent analyses. Specifically, an average $P_P$ value was estimated for each experimental replicate by calculating the geometric mean of the eight segment-specific $P_{P,i}$ values. The arithmetic mean of each of these 13 summary $P_P$ values was 0.58 (mean ± SD = 0.50–0.64). The models described below use the average $P_P$ value of 0.58 for simplicity.

**Predicted costs of IVGs for cellular infectivity.** If singular infections often result in replication of fewer than eight viral gene segments, then multiple particles would be required to productively infect a cell. To evaluate the relationship between the frequency of IVGs and the number of particles required to infect a cell, we developed a probabilistic model in which the likelihood of segment delivery is governed by the parameter $P_P$. In Fig. 2a, we examine how $P_P$ affects the frequency with which a single virion delivers a given number of segments. If $P_P$ is low, singular

segment-specific values to reflect the fact that productive infection requires independent delivery of all eight genome segments.

Using this assay, the $P_{P,i}$ values for each segment of Pan/99 virus were quantified (Fig. 1a). We observed that each segment was present at an intermediate frequency between 0.5 and 0.7, indicating that IVGs may arise from loss of any segment(s).

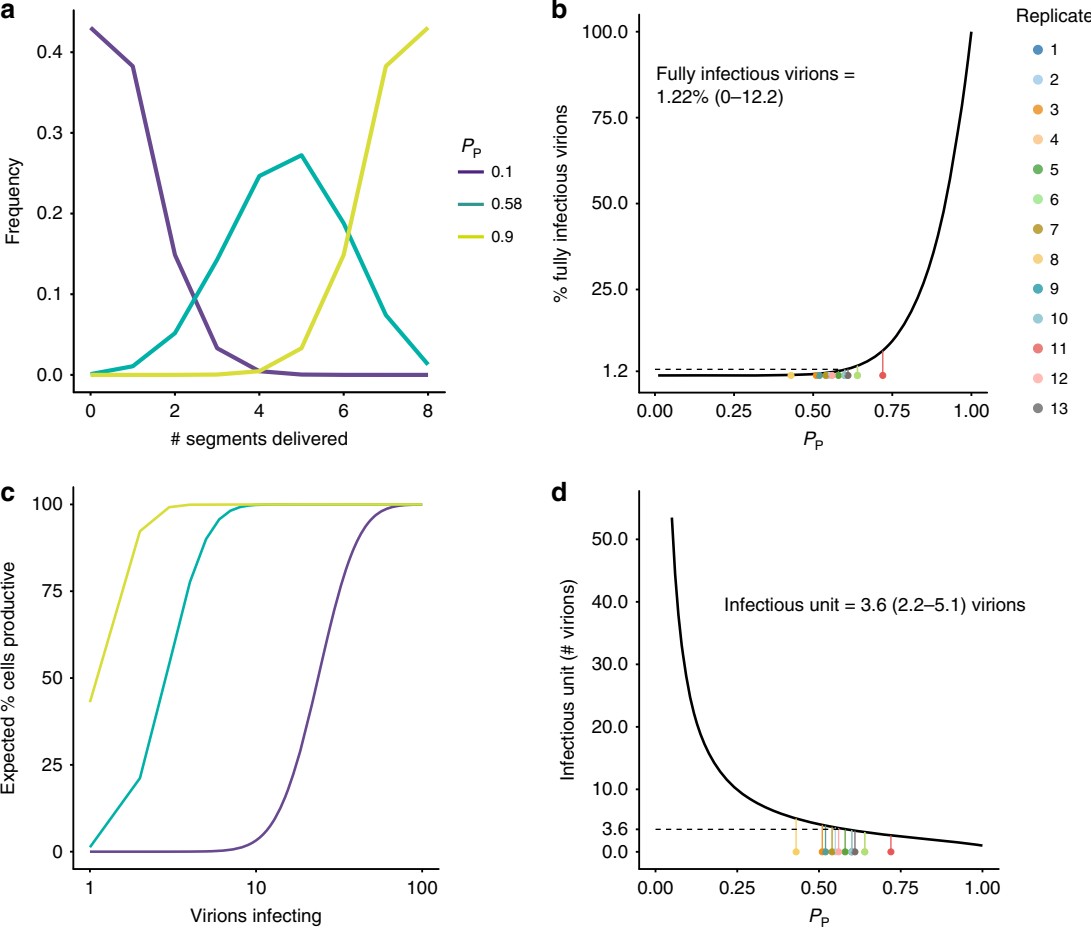

**Fig. 2** Incomplete genomes require complementation for productive infection at the cellular level. **a** The expected number of segments delivered upon infection with a single virion was calculated for two extreme values of $P_P$ (0.10, 0.90) and the estimated $P_P$ of Pan/99 virus (0.58, mean ± SD 0.50–0.64). **b** The percentage of virions expected to initiate productive infection was plotted as a function of $P_P$. Colored points along the bottom of the plot correspond to the average $P_P$ value of each experimental replicate in Fig. 1, with lines connecting them to their predicted value on the theoretical line, and therefore represent predicted frequencies for Pan/99 virus. Mean ± SD (given by Eqs (8) and (9)) interval is given in the text above the line. **c** The percentage of cells expected to be productively infected following infection with a given number of virions was calculated for the same $P_P$ values as in (**a**). **d** The expected number of virions required to make a cell productively infected is plotted as a function of $P_P$. As in (**b**), colored points correspond to the average $P_P$ value of each Pan/99 experimental replicate in Fig. 1, and mean ± SD (given by Eqs (13) and (14)) interval is given in the text. Source data are provided as a Source Data file

infections typically yield few segments per cell. Even at the intermediate $P_P$ that characterizes Pan/99 virus, the vast majority of singular infections give rise to IVGs within the cell. When $P_P$ is high, however, most cells receive the full complement of eight segments. In Fig. 2b, we plot the relationship between $P_P$ and the percentage of cells that are expected to be productively infected following singular infection. If only a single virus infects a cell, then the probability that all eight segments are present will be $P_P^8$. For Pan/99 virus, the frequency with which eight segments are present is $\sim 0.58^8 = 1.22\%$ (mean ± SD = 0–12.2%).

Importantly, however, if more than one virus particle infects the cell, then the probability that all eight segments are present will be considerably higher. This effect is demonstrated in Fig. 2c, where the percentage of cells containing all eight IAV segments is plotted as a function of the number of virions that have entered the cell. Here, we see that, even for low $P_P$, a high probability of productive infection is reached at high multiplicities of infection. Finally, in Fig. 2d, the relationship between $P_P$ and the average number of virions required to productively infect a cell is examined. We see that the number of virions comprising an infectious unit increases sharply at low values of $P_P$. Based on our

experimentally determined values of $P_P$ for Pan/99 virus ($P_P = 0.58$), we estimate that an average of 3.6 (mean ± SD = 2.2–5.1) virions must enter a cell to render it productively infected (Fig. 2d). Thus, as a result of stochastic loss of gene segments, the likelihood that a full viral genome will be replicated within a singularly infected cell is low. The fitness implications of this inefficiency may be offset, however, by complementation of IVGs in multiply infected cells.

**Predicted costs of IVGs for population infectivity**. The potential for multiple infection to mitigate the costs of inefficient genome delivery will, of course, depend on the frequency of multiple infection. To predict the impact of IVGs on viral fitness, we modeled the process of infection at a population level. A given number of virions were delivered to a population of $10^6$ cells. The probability that at least one cell would receive all eight genome segments, and thus the population would become infected, was calculated over a range of MOIs for viruses of different $P_P$ values (Fig. 3a). The resultant plot shows that viruses with lower $P_P$ require markedly higher MOIs to ensure productive infection within a population of cells. When we estimated the MOI

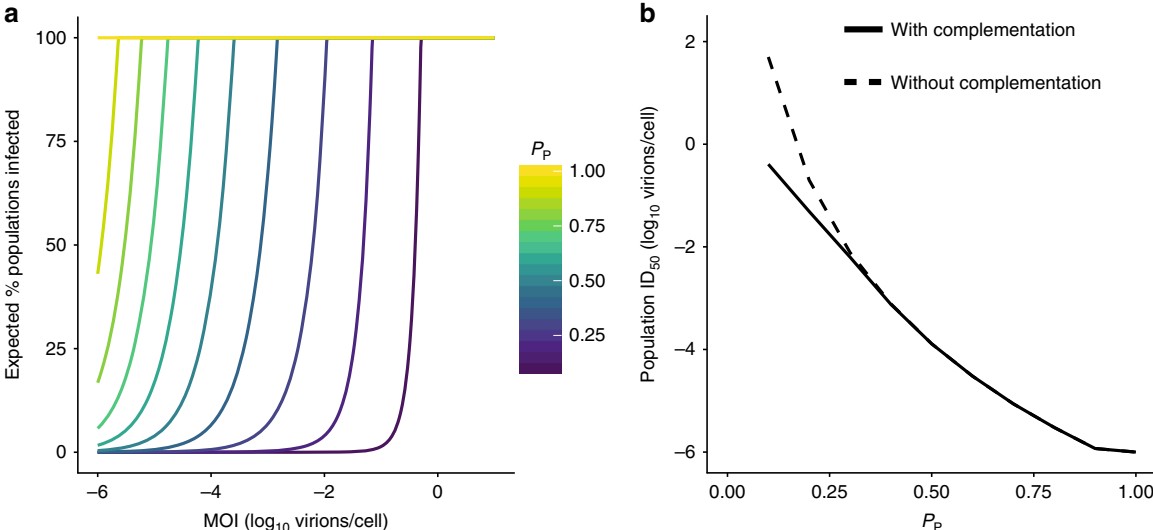

**Fig. 3** Requirement for coinfection poses a barrier to establishing an infection in a population of cells. To define the impact of IVGs on the ability of a virus to establish infection in a population of cells, the probability that a population of $10^6$ cells became infected by a given number of virions was calculated. **a** The percentage chance of at least one cell containing eight genome segments following delivery of virions was calculated for each $P_P$ and across a range of MOIs. **b** The MOI that led to 50% of cell populations becoming infected (ID$_{50}$) was plotted as a function of $P_P$, where complementation was possible (solid line), and where only complete viral genomes could initiate infection (dashed line). IVGs = incomplete viral genomes. Source data are provided as a Source Data file

required for a virus of a given $P_P$ to have a 50% chance of infecting a population, we observed that the ID$_{50}$ increases on a logarithmic scale as $P_P$ decreases (Fig. 3b). To evaluate the contribution of IVGs to the establishment of infection, we estimated the ID$_{50}$ when complementation cannot occur. Here, infecting a population instead requires that at least one virion deliver all eight genome segments. This analysis revealed that ID$_{50}$ is not affected by complementation when $P_P$ is above 0.3, indicating that the frequency of fully infectious particles is the main determinant of infectivity under these conditions. Thus, a reliance on multiple infection in a well-mixed system is predicted to bear a substantial fitness cost, and the establishment of infection is likely driven by the rare minority of fully infectious particles. It is important to note that these calculations depend on the number of cells in the population being considered, and so the probability of establishing infection may be influenced by the surface area of target tissue.

**Model of spatially structured viral spread**. The above estimates of viral infectivity assume that virus is distributed randomly over a population of cells. Following the initial infection event, however, viruses spread with the spatial structure. We hypothesized that this structure may be very important for reducing the costs of genome incompleteness once infection is established. To test this idea, we developed a model of viral spread in which the degree of the spatial structure could vary.

The system comprises a spatially explicit grid of cells that can become infected with virus. The number and type of segments delivered upon infection are dictated by the parameter $P_P$ and, if all eight segments are present, a cell produces virions. These virions can then diffuse in a random direction, with the distance traveled governed by the diffusion coefficient ($D$). $D$ was varied in the model to modulate the spatial structure of viral spread: higher $D$ corresponds to greater dispersal of virus, and therefore lower spatial structure. We simulated replication of two virus strains under a range of diffusion coefficients, one with a frequency of IVGs characteristic of Pan/99 virus ($P_P = 0.58$) and one with complete genomes ($P_P = 1.0$).

Our results point to an important role for the spatial structure in determining the efficiency of infection. When $P_p = 1.0$, replication proceeds faster at higher values of $D$, because virus particles reach permissive cells more efficiently (Fig. 4a; Supplementary Fig. 2A–D). In contrast, when $P_p = 0.58$, replication proceeds fastest at intermediate values of $D$ (Fig. 4a; Supplementary Fig. 2E–H). An intermediate level of spatial structure is optimal for a virus with incomplete genomes for two reasons. At high values of $D$, virions diffuse farther and cellular coinfection becomes less likely, reducing the likelihood of complementation. At the other extreme, when $D$ is very low, complementation occurs readily but spread to new cells becomes rare. Note, values of $D > 10^3$ μm$^2$/s approximate even mixing.

The model allows the potential costs of incomplete genomes to be evaluated by comparing results obtained for a virus with $P_P = 1.0$ to those obtained for a virus with a lower $P_P$. In particular, we focused on $P_P = 0.58$ based on the measured values for Pan/99 virus. Thus, in Fig. 4b–d, we evaluated three different measures of viral fitness and plotted the relative values for a virus with $P_P = 0.58$ compared with a virus with $P_P = 1.0$. We show that reductions in the initial growth rate (Fig. 4b) and increases in the time taken to productively infect 100 cells or produce $10^5$ virions (Fig. 4c, d, respectively) brought about by IVGs vary with the spatial structure. We see that the costs of IVGs are minimized at intermediate values of $D$. The greatest costs are seen at higher values of $D$, as the process of virion dispersal approximates random mixing. The results of this model thus predict that the spatial structure of viral spread influences the fitness costs of IVGs.

**Impact of MOI on efficiency of virus production**. Burst size, the average number of virions generated by an infected cell, is an important factor determining the potential for complementation of incomplete genomes. If an infected cell produces a larger number of viral progeny, the likelihood of coinfection in neighboring cells increases. We therefore measured this parameter experimentally for Pan/99 virus by performing single-cycle growth assays over a range of MOIs (1, 3, 6, 10, and 20 PFU/cell).

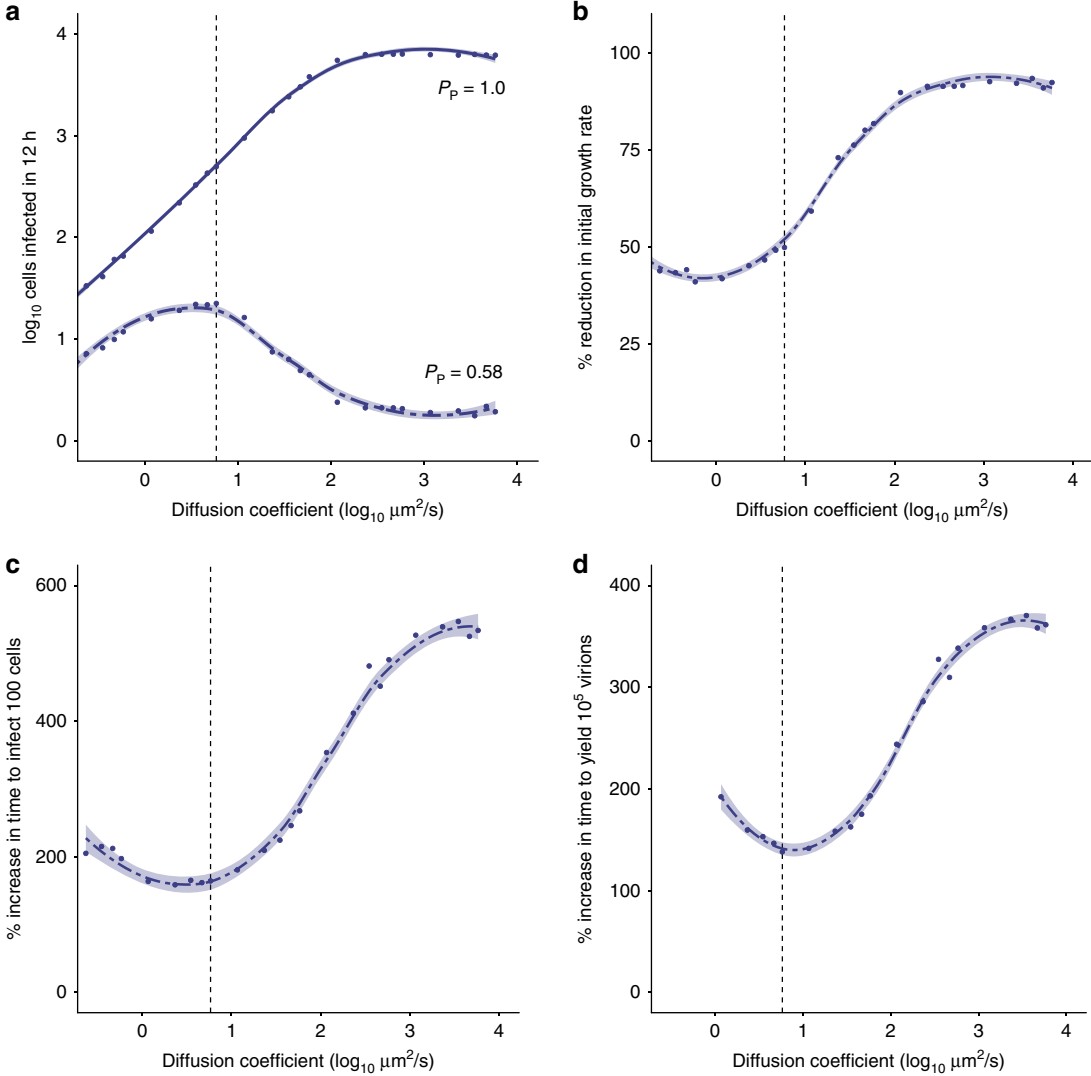

**Fig. 4** The fitness costs of incomplete genomes may be mitigated by spatially structured spread. The dynamics of multi-cycle replication in a 100 × 100 grid of cells were simulated, starting from a single cell in the center of the grid. **a** The initial growth rate (estimated by the log-transformed number of cells that are productively infected in the first 12 h) is shown across a range of diffusion coefficients for a virus with $P_P = 1.0$ (solid line) and $P_P = 0.58$ (dashed line). **b–d** The fitness cost of IVGs, as measured by the reduction in initial growth rate (**b**) or the increase in time taken to infect 100 cells (**c**) and produce $10^5$ virions (**d**), is shown across a range of diffusion coefficients. The vertical dashed line represents the estimated value of $D$ (5.825 μm²/s) for a spherical IAV particle in water. Each point shows the mean of ten simulations. Curves were generated by local regression. Shading represents 95% CI of local regression (mean ± 1.96 * SE). IVGs = incomplete viral genomes. Source data are provided as a Source Data file

Multiple MOIs were used to determine whether burst size is dependent on the number of viral genome copies per cell. We saw that higher MOIs resulted in earlier emergence of virus, suggesting that there is a kinetic benefit of additional vRNA input beyond what is required to productively infect a cell (Fig. 5a; Supplementary Fig. 3). Despite these kinetic benefits, MOIs above 3 PFU/cell conferred no benefit in terms of percent infection (Fig. 5b) or total productivity (Fig. 5c). This growth analysis indicated that a maximum of 11.5 PFU per cell (mean ± SD = 8.1–14.9 PFU/cell) was produced during Pan/99 virus infection of MDCK cells. Based on measured $P_P$ values, these data estimate that a single productively infected cell produces 962 virions, and this value was used as the burst size in our models.

**Impact of MOI and spatial structure on IVG complementation.** Our models indicate that, for a virus of a given $P_P$, the frequency of infected cells containing IVGs is reduced (i) at higher MOIs and (ii) under conditions of high spatial structure. These

predictions can be seen in Fig. 6. In Fig. 6a, we show how the proportion of infected cells that lack a complete viral genome is predicted to vary with MOI under single-cycle conditions. In Fig. 6b, we show how this proportion varies with the structure imposed by diffusion during multi-cycle replication (Fig. 6b). We tested the predictions of these models experimentally by modulating MOI and spatial spread in IAV-infected cell cultures and gauging the impact of each manipulation on levels of IVGs.

First, to evaluate the spatial structure under single and multi-cycle conditions, we used a reporter strain of Pan/99 virus with a tetracysteine tag on the NP protein (Pan/99-NP_TC virus) and visualized infected cell monolayers. The results confirmed that single-cycle inoculation results in random dispersal of virus across cells, while multi-cycle replication proceeds in a spatially structured manner resulting in foci of infection (Fig. 6c).

Next, we set up an experiment using Pan/99-WT and Pan/99-Helper viruses that lack a TC tag, but carry different epitope tags fused to their HA proteins. To monitor levels of IVGs, we used

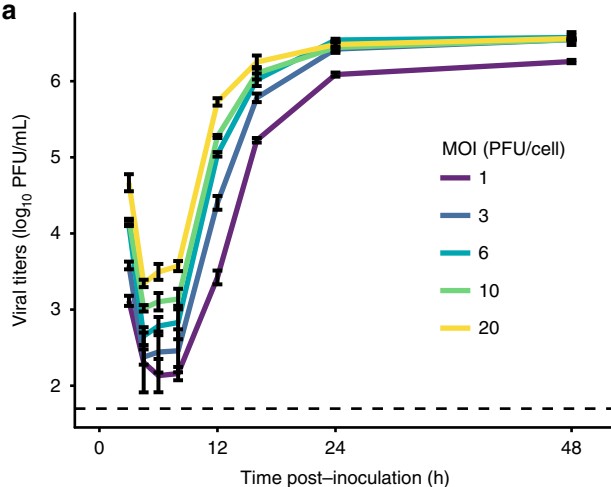

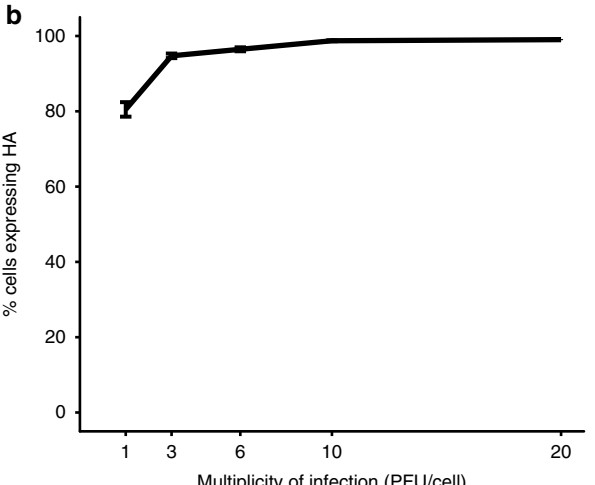

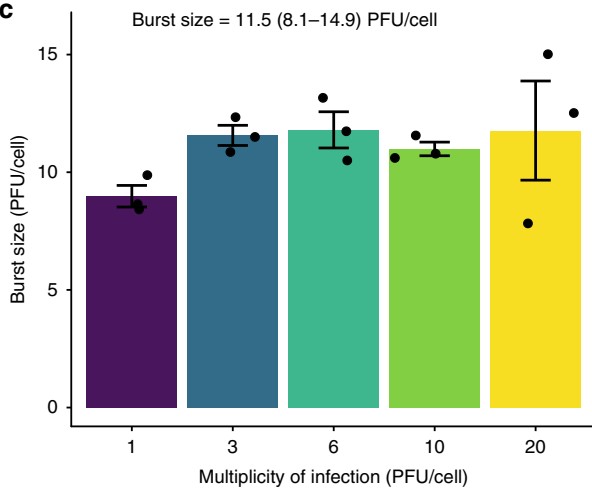

Burst size = 11.5 (8.1–14.9) PFU/cell

**Fig. 5** Burst size of Pan/99 virus is constant over a range of high MOIs. **a** MDCK cells were inoculated with Pan/99-WT virus at MOIs of 1, 3, 6, 10, and 20 PFU/cell under single-cycle conditions. Infectious titers at each time point are shown, with MOI indicated by the colors defined in the legend. Dashed line indicates the limit of detection (50 PFU/mL). **b** Fraction of cells expressing HA at each MOI, as measured by flow cytometry staining of cells 12 h post inoculation. **c** Burst size in PFU produced per HA+ cell, with individual data points overlaid on bars. Text indicates mean ± SD estimate of burst size. In all panels, mean and standard error ($N = 3$ replicates per MOI) are plotted, and colors correspond to the legend in panel **a**. Source data are provided as a Source Data file

flow cytometry targeting these epitope tags to measure the potential for complementation—that is, the benefit provided by the addition of Pan/99-Helper virus. We hypothesized that, under single-cycle conditions, the potential for complementation would decrease with increasing WT virus MOI, since complementation between coinfecting WT viruses would occur frequently at high MOIs. In addition, under multi-cycle conditions initiated from low MOI, we predicted that the potential for complementation would be greatest at the beginning of infection, due to the random distribution of viral particles, and reduced by secondary spread. We hypothesized that the combination of local dispersal and high particle production during secondary spread would support coinfection in neighboring cells. To test our hypotheses, we inoculated cells with Pan/99-WT virus and either added Pan/99-Helper virus at the same time, or added the Helper virus after allowing time for secondary spread.

To evaluate the potential for complementation at the outset of infection and at a range of MOIs, cells were coinoculated with Pan/99-Helper virus at a constant MOI and with Pan/99-WT virus at MOIs of 0.1, 0.3, 0.6, or 1 PFU/cell. Cells were then incubated under single-cycle conditions for 12 h to allow time for HA protein expression. Samples were processed by flow cytometry with staining for WT and Helper HA proteins (Supplementary Fig. 4). In each coinfection, we quantified the benefit provided by Pan/99-Helper virus by calculating the enrichment of WT HA expression in Helper+ cells relative to Helper− cells. Essentially, the enrichment measure works as follows. If the proportion of Helper+ cells that are WT+ is higher than the proportion of Helper− cells that are WT+, enrichment will be >0%, indicating a cooperative interaction in which Helper virus allows the expression of WT HA genes present in incompletely infected cells. The results shown in Fig. 6d revealed that the potential for complementation at the outset of infection was high at low MOIs, but decreased with increasing MOI. This result was as expected, since complementation between WT virus particles was predicted to reduce the need for Helper virus (Fig. 6a).

To evaluate the impact of spatially structured secondary spread on IVG prevalence, cells were inoculated with Pan/99-WT virus at low MOI (0.002 or 0.01 PFU/cell) and then multi-cycle replication was allowed to proceed over a 12 h period. After this period, cells were inoculated with Pan/99-Helper virus to complement any incomplete genomes, and incubated for 12 h under single-cycle conditions to allow HA expression to occur. In contrast to the results seen when complementation was offered at the outset of infection, the enrichment of WT+ cells in the Helper+ fraction was significantly lower in these samples where multi-cycle replication occurred prior to the addition of Helper virus. This reduction in enrichment is clear when comparing infections performed under each condition in which ~50% of cells expressed WT HA (Fig. 6d). These data agree with our theoretical results (Fig. 6b) and indicate that the spatial structure of secondary spread facilitates complementation between WT particles as they infect neighboring cells at locally high MOIs.

To account for the possibility that spread of the WT virus over 12 h would reduce the potential for complementation by super-infection exclusion, we analyzed the level of Helper virus infection in relation to the presence of WT virus (Supplementary Fig. 4D). We observed that a similar fraction of cells expressed Helper HA in (i) infections with Helper virus alone (data plotted at 0% WT HA+), (ii) simultaneous coinfections of WT and

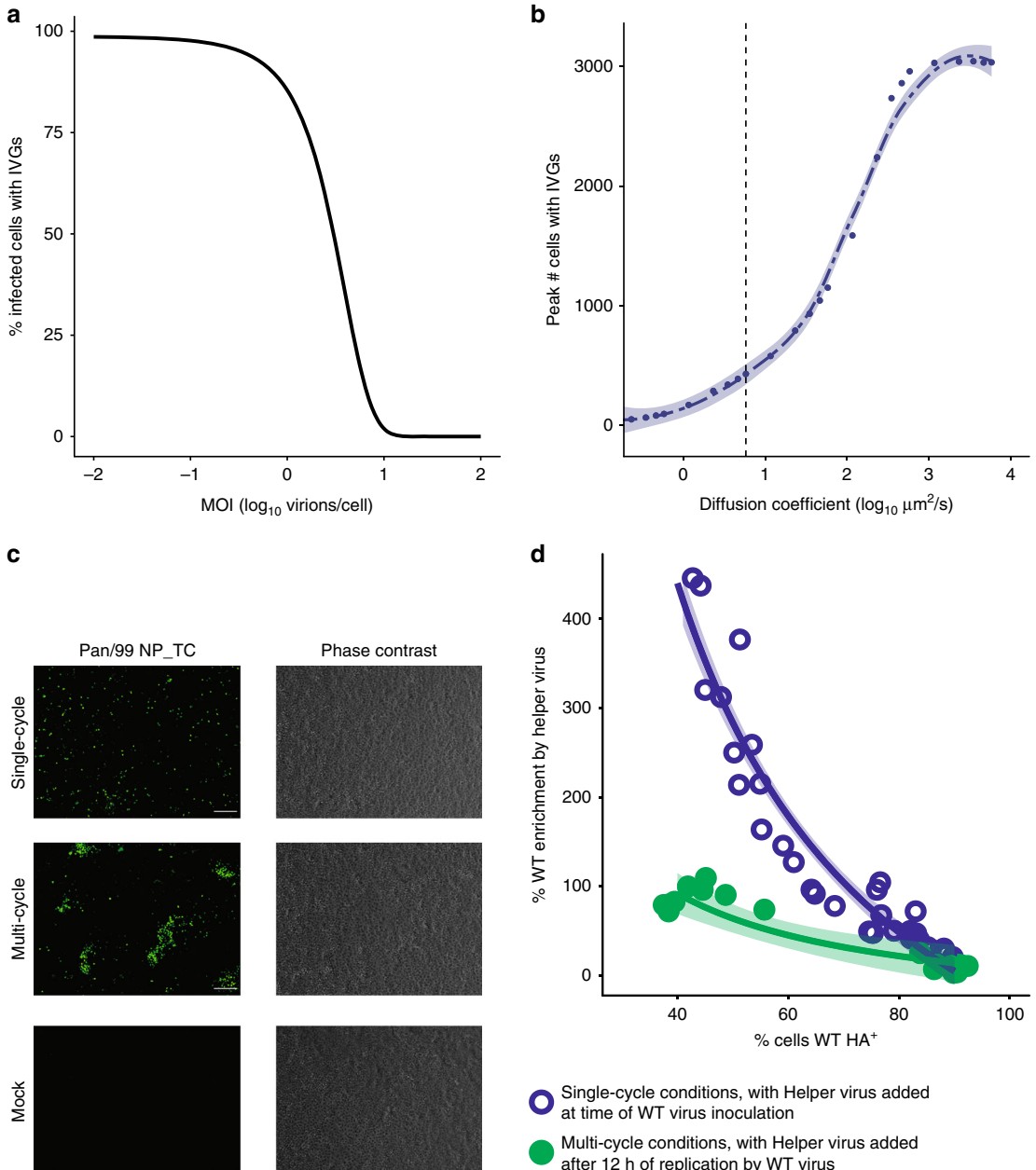

**Fig. 6** Complementation of incomplete genomes occurs efficiently at high MOI and during secondary spread from low MOI. **a** The model shown in Fig. 3 was used to calculate the probability that a given infected cell contained an IVG. The percentage of infected cells that contain fewer than eight segments is shown at a range of MOIs for $P_P = 0.58$. **b** An infection in which multi-cycle replication occurs with the spatial structure was simulated as in Fig. 4. The maximum number of cells that contain IVGs in each simulation is shown for a range of diffusion coefficients. Shading represents 95% CI of local regression (mean ± 1.96 * SE). **c** Visualization of spatially structured and unstructured infections. Cells were inoculated with Pan/99-NP_TC virus and incubated under single-cycle conditions for 12 h (top), multi-cycle conditions for 12 h followed by single-cycle conditions for 12 h (middle), or sham-inoculated and incubated under single-cycle conditions for 12 h (bottom), then visualized by FlaSH staining (left) or phase-contrast imaging (right). Scale bar represents 200 μm. **d** The extent to which the presence of Pan/99-Helper virus increased WT HA positivity (% enrichment) was evaluated at the outset of infection (open blue circles) and following secondary spread (filled green circles). To gauge potential for complementation at the outset of infection, cells were simultaneously inoculated with Pan/99-WT virus and Pan/99-Helper, then incubated under single-cycle conditions for 12 h ($N = 3$ replicates per MOI, four MOIs per experiment, and three independent experiments). To test the impact of secondary spread on potential for complementation, cells were inoculated with Pan/99-WT virus at low MOI and incubated under multi-cycle conditions for 12 h, then inoculated with Pan/99-Helper virus and incubated under single-cycle conditions for 12 h ($N = 3$ replicates per MOI, two MOIs per experiment, and three independent experiments). Curves represent estimates of a fixed effects model with the formula $\%\text{Enrichment} = \beta_1 \frac{\text{Multi−cycle}}{\%\text{WT HA}^+} + \beta_2 \frac{1}{\%\text{WT HA}^+} + \beta_3^*\text{Multi − cycle}$, with shading representing 95% CI (mean ± 1.96 * SE) of model estimate. IVGs = incomplete viral genomes. Source data are provided as a Source Data file

Helper viruses, and (iii) infections where Helper virus was added 12 h after low MOI WT virus. If superinfection exclusion was limiting the ability of Pan/99-Helper virus to complement IVGs among cells, we would expect lower frequencies of Helper HA

expression in populations where Pan/99-Helper virus was added after WT virus. Although superinfection exclusion develops in less than 12 h[8], this outcome was expected because, under the multi-cycle conditions used, the vast majority of infected cells

present at 12 h were recently infected in a second or third round of multiplication.

**A virus with absolute dependence on multiple infection**. To evaluate the potential for complementation in vivo, we generated a virus that is fully dependent on complementation for replication. This was accomplished by modifying the M segment to generate one M segment which encoded only M1 (M1.Only), and a second one which encoded only M2 (M2.Only) (Fig. 7a). When combined with seven standard reverse genetics plasmids for the remaining viral gene segments, the plasmids encoding these two M segments allowed the generation of a virus population in which individual viruses encode functional M1 or M2, but not both. We called this virus Pan/99-M.STOP virus. Due to the rarity of recombination within segments in negative-sense RNA viruses[32], it is unexpected that M1.Only and M2.Only segments will recombine to generate a WT M segment. Hence, this virus is reliant on both M segments being delivered to the same cell by coinfection. It is important to note that, in contrast to the more arbitrary multiplicity dependence of a wild-type IAV, the complementation needed by Pan/99-M.STOP virus requires coinfection with two viruses of a particular genotype.

To characterize the Pan/99-M.STOP virus genetically, we used digital droplet PCR (ddPCR) to measure copy numbers of the two M segments and the NS segment in three virus samples (Fig. 7b). The total M segment copy number was found to comprise 30% M2.Only and 70% M1.Only. In addition, the total number of M segments was similar to the number of NS segments, as expected if each virion packages one NS and one M vRNA (Fig. 7b). To verify that M1.Only and M2.Only M segments were packaged into distinct virions, we performed infections of MDCK cells with serial dilutions of Pan/99-M.STOP virus under single-cycle conditions and analyzed expression of M1 and M2 by flow cytometry. We observed that, as dilution increased, cells expressing M1 were less likely to express M2, and vice versa (Fig. 7c). This result would be expected if expression of both proteins from the same cell required coinfection with M1.Only and M2.Only encoding virions. As a control, we monitored the effect of dilution on co-expression of HA and M1 or M2. Here, we found that co-expression of M1 or M2 and HA was much less sensitive to dilution, consistent with co-delivery of M and HA segments by single virions. At limiting dilutions, where only 2% of cells were infected, only one sample reached the point of absolutely no co-expression between M1 and M2, suggesting that stocks may contain some aggregates of virus particles comprising fully infectious units. We cannot fully exclude this possibility, but note that co-expressing cells represented <10% of infected cells at this limiting dilution, suggesting that most co-expression of M1 and M2 at higher concentrations of virus is indeed mediated by coinfection.

M2 expression was shown previously to be nonessential for replication in cell culture[33,34]. To evaluate the extent to which Pan/99 virus relies on M2 expression for viral multiplication in the systems used here, we generated a virus unable to express M2 using the plasmid encoding the M1.Only segment. This virus was successfully recovered and formed small plaques in MDCK cells. Multiple attempts to culture the M1.Only virus in MDCK cells and eggs failed, however, indicating the importance of M2 to viral propagation in these substrates.

**Consequences of multiplicity dependence for infectivity**. To test the hypothesis that a given number of Pan/99-M.STOP virus particles would be less infectious than a comparable number of Pan/99-WT virus particles, we characterized both viruses using a series of titration methods that vary in their dependence on

infectivity and M protein expression. We first used ddPCR to quantify NS copy numbers of the WT and M.STOP viruses and then normalized all other comparisons with this ratio to account for the difference in virus concentration. As shown above, total M copy numbers were roughly equivalent when normalized to NS (Fig. 7b, d). Using immunotitration, in which cells are infected under single-cycle conditions with serial dilutions of virus and then stained for HA expression[35], we observed equivalent titers of both viruses (Fig. 7d). This was expected, as HA expression under single-cycle conditions is not dependent on M1 or M2 proteins. When titration relied upon multi-cycle replication, however, the WT virus was higher titer than the M.STOP virus. This difference was moderate in cell culture-based measurements, with PFU and $TCID_{50}$ titers 24- and 51-fold higher, respectively, likely because of the reduced importance of M2 in this environment. The full cost for infectivity of separating the M1 and M2 ORFs onto distinct segments was apparent in vivo, where 815 times as much M.STOP virus was required to infect 50% of guinea pigs compared with WT virus (Fig. 7d). Thus, although the M.STOP virus differs from a virus with very low $P_P$ in that complementation can only occur when viruses carrying M1.Only and M2.Only segments coinfect, the prediction shown in Fig. 3, that increased dependence on multiple infection decreases infectivity, held true in this system.

**Potential for complementation in vivo**. Having determined that the dependence of Pan/99-M.STOP virus on complementation impairs viral infectivity, we next sought to evaluate the potential for complementation to occur in vivo once infection had been established. Guinea pigs were inoculated intranasally with equivalent doses of Pan/99-WT or Pan/99-M.STOP virus in terms of NS vRNA copies. Specifically, a dose of $10^7$ copies per guinea pig was used to ensure successful Pan/99-M.STOP virus infection in all animals. This dose represents $8 \times GPID_{50}$ of this mutant virus, and $6.5 \times 10^3 \times GPID_{50}$ of the WT virus. Despite its reduced ability to establish infection, Pan/99-M.STOP virus successfully grew in guinea pigs, following similar kinetics to Pan/99-WT virus. Average peak virus production, measured as NS vRNA copies, was reduced by only ninefold relative to WT (Fig. 8a). Sanger sequencing of viral cDNA from nasal washes confirmed that both M1.Only and M2.Only segments were present in vivo, and quantification of the the two alleles by ddPCR revealed a bias toward M1.Only segments (Supplementary Fig. 5). Because the inoculum of WT and M.STOP viruses comprised the same dose in terms of vRNA copies, but different doses in terms of $GPID_{50}$, we also inoculated guinea pigs with WT virus at doses of $8 \times GPID_{50}$ and $6.5 \times 10^3 \times GPID_{50}$ (Fig. 8b). This experiment was designed to define the contribution of the effective dose to the differences observed between M.STOP and WT viruses in guinea pigs. Similar peak titers and kinetics of shedding were observed in both groups of WT virus infected guinea pigs, indicating minimal dose dependency (Fig. 8b).

Finally, we conducted an experiment to determine whether Pan/99-M.STOP virus was competent to undergo transmission to new hosts. In this case, guinea pigs were inoculated with equivalent doses in terms of $GPID_{50}$ with the goal of establishing comparable infections in the donor hosts so that relative efficiency of transmission could be better evaluated. Thus, doses of $8 \times GPID_{50}$ of WT or M.STOP virus were used. At 24 h post inoculation, each index guinea pig was co-housed with a naive partner. As expected, WT virus transmitted to and initiated robust infection in each of the four contact animals. In contrast, only transient, low levels of the M.STOP virus were observed in nasal washings collected from contacts (Fig. 8c). These results suggest that the spatial structure inherent to multi-cycle

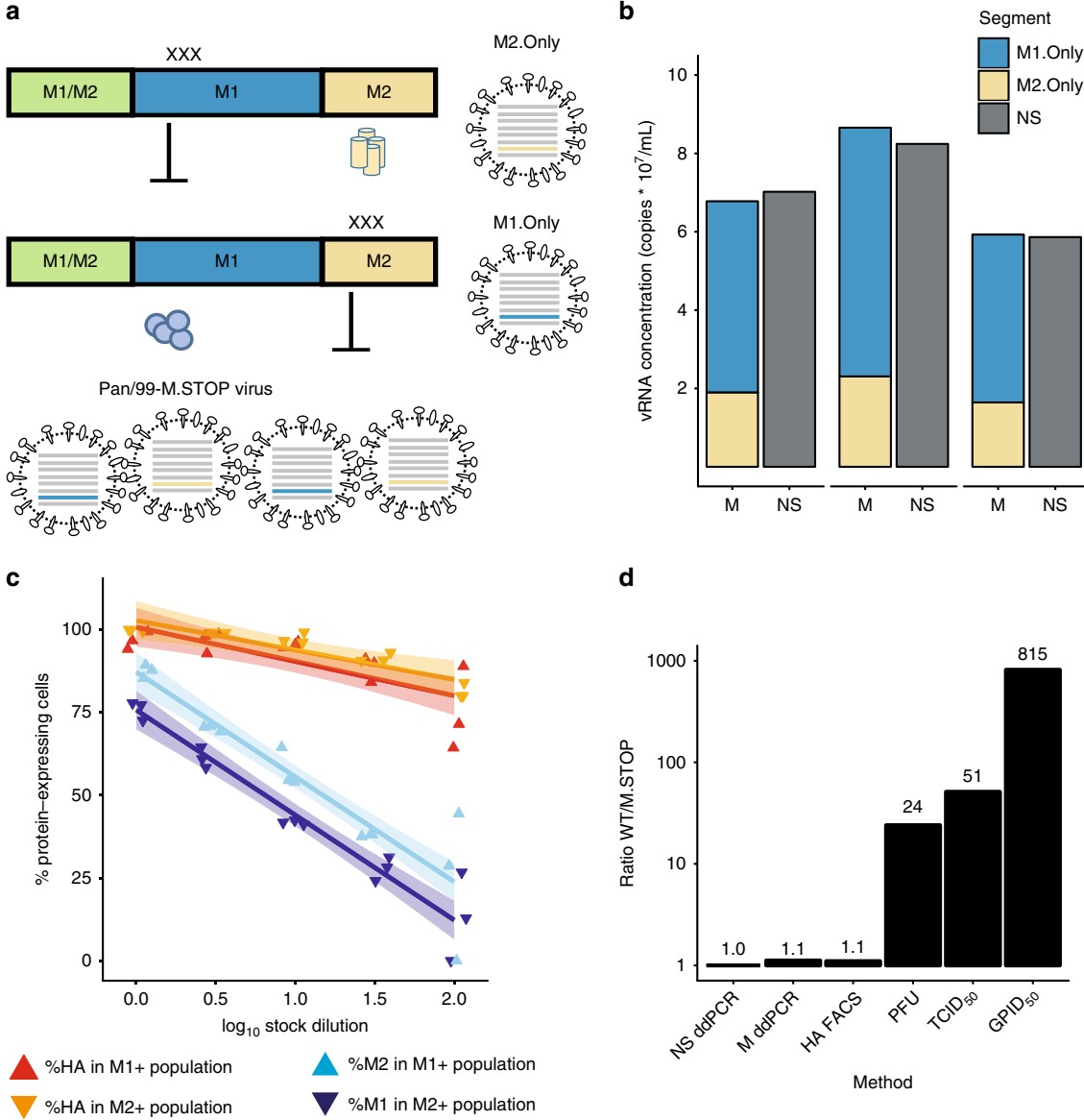

**Fig. 7** Dependence on complementation hinders viral infectivity. **a** Mutation scheme used to generate M1.Only and M2.Only segments, and Pan/99-M.STOP virus. **b** Copies of M1.Only, M2.Only, and NS segments in three separate aliquots of Pan/99-M.STOP virus stock were quantified by digital droplet PCR. **c** Cells were inoculated with Pan/99-M.STOP virus, and incubated under single-cycle conditions before staining for HA, M1, and M2 expression ($N = 3$ replicates per dilution). The percentage of cells expressing M1, M2, and HA within M1$^+$ or M2$^+$ subpopulations is shown at each dilution. Lines represent linear regression with shading representing 95% CI (mean $\pm$ 1.96 * SE). **d** Titers of WT and M.STOP virus stocks were quantified by ddPCR targeting the NS segment ($N = 2$ (WT) or 3 (M.STOP) replicates), ddPCR targeting (any) M segment ($N = 2$ (WT) or 3 (M.STOP) replicates), immunotitration by flow cytometry ($N = 1$ replicate per virus per dilution), plaque assay ($N = 6$ replicates per virus), tissue culture ID$_{50}$ ($N = 4$ replicates per virus per dilution), and guinea pig ID$_{50}$ ($N = 4$ animals per virus per dose). All results are normalized to the ratio of NS ddPCR copy numbers. Source data are provided as a Source Data file

replication mitigates the cost of incomplete genomes in an individual host, but that dependence on complementation is costly for transmission. This result is consistent with the theoretical predictions displayed in Figs. 2–4.

## Discussion

Using a single-cell approach that enables detection of incomplete IAV genomes, we show that ~99% of Pan/99 virus infections led to replication of fewer than eight segments. The theoretical models we describe predict that the existence of IVGs presents a need for cellular coinfection, and that this need has a high probability of being met when spread occurs in a spatially

structured manner. Use of silent genetic tags allowed us to experimentally interrogate cooperation at the cellular level to test these predictions. Cell culture experiments showed that coinfection and complementation occur readily when multiple rounds of infection are allowed to proceed with spatial structure. The high potential for complementation to occur in vivo was furthermore revealed by the robust within-host spread of a virus that is fully dependent on coinfection. Complementation was not observed during transmission; however, suggesting that fully infectious particles may be required to initiate infection in a new host.

The existence of incomplete genomes was previously predicted by Heldt et al., and these predictions are consistent with the experimental findings of our single-cell assay[21]. The parameter

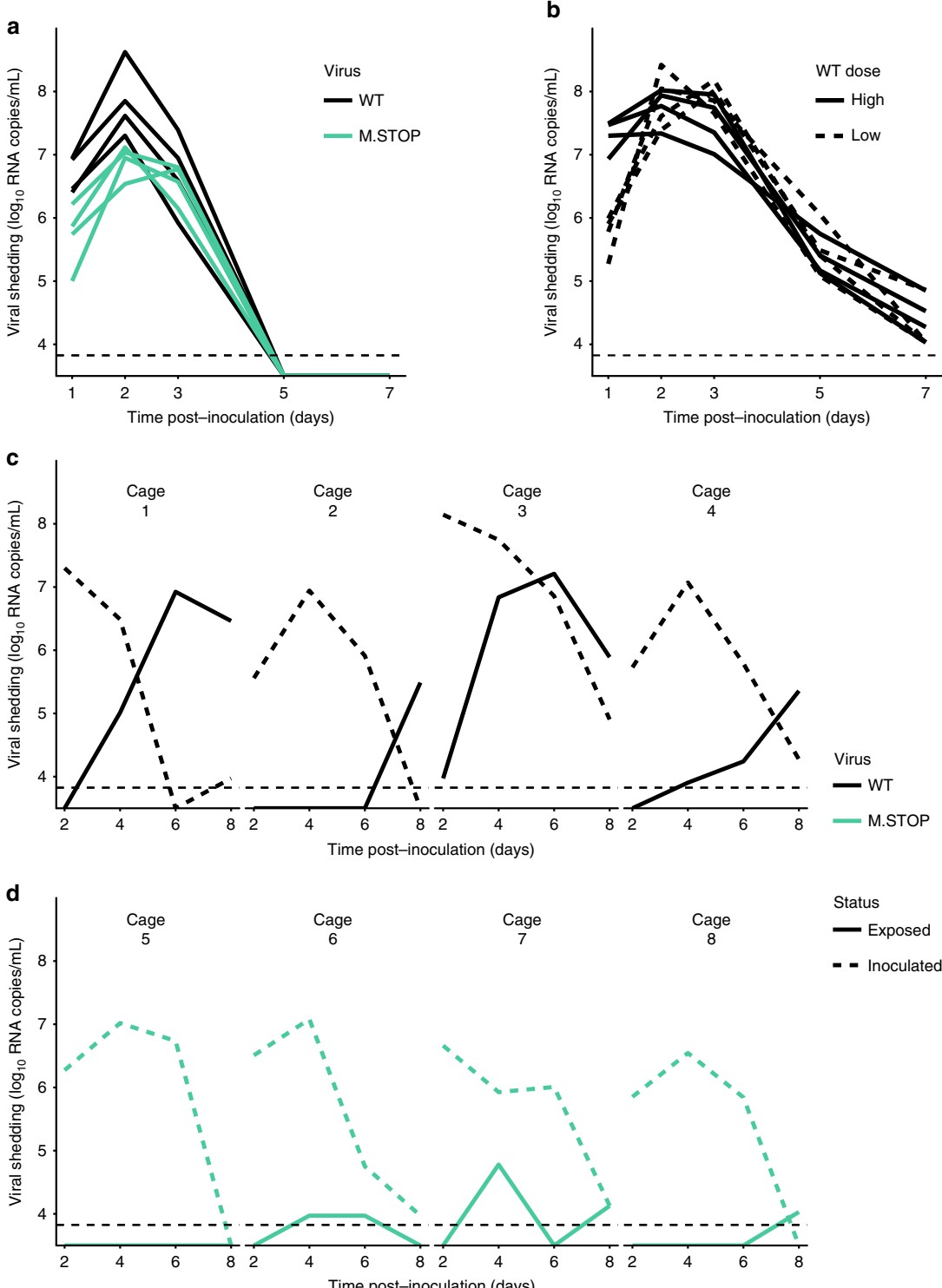

**Fig. 8** Dependence on complementation hinders viral transmission, but has a more modest effect on replication. **a** Guinea pigs were inoculated with $10^7$ RNA copies of Pan/99-WT virus or Pan/99-M.STOP virus, and nasal washes were collected over 7 days to monitor shedding. NS segment copy number per mL of nasal lavage fluid is plotted ($N = 4$ animals). **b** Guinea pigs were inoculated with $10^7$ or $1.23 \times 10^4$ RNA copies of Pan/99-WT virus, corresponding to $6.5 \times 10^3$ and eight guinea pig infectious dose ($GPID_{50}$), respectively. Nasal washes were collected over 7 days to monitor shedding. NS segment copy number per mL of nasal lavage fluid is plotted for high (solid lines) and low (dashed lines) doses ($N = 4$ animals per group). **c**, **d** Guinea pigs were inoculated with $8 \times GPID_{50}$ of Pan/99-WT virus (**c**) or Pan/99-M.STOP virus (**d**) and co-housed with uninfected partners after 24 h. Nasal washes were collected over the course of 8 days to monitor shedding kinetics and transmission between cagemates. NS segment copy number per mL of nasal lavage fluid is plotted ($N = 2$ animals per cage, four cages per group). In all plots, horizontal dotted lines represents the limit of detection (6696 RNA copies/mL). Source data are provided as a Source Data file

estimated by this assay, $P_P$, is defined as the probability that, following infection with a single virion, a given genome segment is successfully replicated. Previous work by Brooke et al. showed that cells infected at low MOIs express only a subset of viral proteins[20]. While lack of protein expression could be explained by a failure in transcription or translation, the results of our single-cell sorting assay indicate that the vRNA segments themselves are absent, as they should be amplified by the helper virus polymerase even if they do not encode functional proteins. As in Brooke et al., our method does not discriminate between the alternative possibilities that segments are absent from virions themselves or are lost within the cell, but published results suggest that individual virions usually contain a full genome[24,25]. Importantly, our single-cell assay quantifies the frequencies of all eight segments, rather than only those encoding proteins that can be stained, and therefore allows for analysis of associations between segments. Despite the importance of interactions among vRNP segments during virion assembly[10,28–31], we did not detect compelling evidence of segment co-occurrence at the level of vRNA replication within target cells. This observation suggests that interactions among segments formed during assembly are likely not maintained throughout the early stages of the viral life cycle. While our measured $P_P$ values are comparable for all segments, independent delivery of segments with distinct probabilities could serve as a mechanism to control gene dosage, similar to what has been observed in multipartite viruses[36]. Indeed, experiments with Pan/99-M.STOP virus showed a consistent bias toward M1-encoding segments in vivo when expression of M1 and M2 was decoupled. This bias may be important for maintaining low M2 protein levels, which we recently found to be an important predictor of viral fitness[37].

The results of our single-cell assay indicate that 1.22% (95% CI 0–12.2%) of Pan/99 virions are fully infectious, which is consistent with our prior estimates based on observed levels of reassortment between Pan/99 wild-type and variant viruses[26]. If these 1.22% of virions comprise the PFUs present in a virus population, then the total number of virions present in a population is equal to 82 times the number of PFUs (because $1/0.0122 = 82$). This result of 1.22% is, however, lower than other reported estimates of the frequency of fully infectious particles. This difference may stem from our use of a helper virus, which likely allows more robust detection of IVGs than would be expected in a system dependent on the detection of nonreplicating viral genomes or their mRNA transcripts[21,22]. In addition, quantitative differences among published reports may relate to use of different virus strains, as Brooke et al. observed that the frequency of IVGs is strain-specific[20]. Interestingly, the strain-specificity of $P_P$ is likely to influence the relative representation of genome segments when two strains of varying $P_P$ reassort during coinfection. As lower $P_P$ values result in successful replication of fewer segments per cell, the genomes of reassortant progeny are likely to contain more segments from the virus with higher $P_P$.

Replication and secondary spread in an individual host involves inherent spatial structure, as virions emerge from an infected cell and travel some distance before infecting a new cell[38–40]. Localized spread is predicted to affect viral population dynamics in multiple ways, including by facilitating abundant coinfection, which we explored herein[39]. We determined that a trade-off exists between complementation and dispersal. Handel et al. explore a similar trade-off related to attachment rates in unstructured populations, finding that an intermediate level of stickiness is optimal—virions that bind too tightly are slow to leave the producing cell, while those that bind too weakly are unable to infect new cells[41]. We observe a similar effect with spatial structure: virions that diffuse farther do not receive sufficient complementation, but not diffusing far enough results in

cells being infected with more virions than necessary, limiting spread. The optimal level of spatial structure is thus an intermediate one that allows virions to efficiently reach new cells while ensuring sufficient complementation.

In quantitative terms, our model predicts that a diffusion coefficient characteristic of a sphere of 100 -nm diameter in water provides a near-optimal level of spatial structure. This may approximate diffusion in cell culture, but the extracellular environment experienced by a virus in vivo would be different. Namely, virus replicating within the respiratory tract would be released into a layer of watery periciliary fluid, which underlies a more viscous mucous blanket[42]. The structure and composition of this epithelial lining fluid may act to limit dispersal of virus particles relative to that expected in cell culture. Importantly, however, this fluid lining the airways is not static, but is moved in a directional manner by coordinated ciliary action[42]. This coordinated movement raises the possibility that IAVs may have evolved to depend upon ciliary action to mediate directional dispersal of virions to new target cells, while maintaining a high potential for complementation of IVGs. This concept will be explored in subsequent studies.

Our experiments designed to test the predicted role of spatial structure in enabling complementation confirmed that secondary spread allows Pan/99-WT virus to replicate efficiently even at low initial MOIs, diminishing the need for complementation after only 12 h of multi-cycle replication. The potential for complementation of IVGs in vivo was furthermore evidenced by the replication in guinea pigs of Pan/99-M.STOP virus, which requires coinfection for productive infection. Importantly, however, Pan/99-M.STOP virus did not transmit to exposed cagemates. It is important to note that productive infection by Pan/99-M.STOP virus requires coinfection with two viruses of a particular genotype, which occurs less frequently than the complementation needed for completion of a WT IAV genome. Despite this caveat, the failure of Pan/99 M.STOP virus to transmit suggests that the establishment of IAV infection requires at least some fully infectious virions. The delivery of multiple particles to a small area via droplet transmission may allow multiple virions to infect the same cell, but our data suggest that this mechanism does not occur efficiently in a guinea pig model. The tight genetic bottleneck observed in human–human transmission events is furthermore consistent with a model in which infection is commonly initiated by single particles[43]. In prior work, mutations decreasing the frequency of fully infectious particles, but not eliminating them entirely, were observed to increase transmissibility[27]. This enhanced transmission was attributed to modulation of the HA:NA balance, which enhanced growth in the respiratory tract. In contrast, the HA:NA balance of Pan/99-M.STOP virus is not expected to differ from Pan/99 virus.

In summary, our findings suggest that incomplete genomes are a prominent feature of IAV infection. These semi-infectious particles are less able to initiate infections in cell culture and during transmission to new hosts, when virions are randomly distributed. In contrast, our data show that incomplete genomes actively participate in the within-host dynamics of infection. Within a host, IVGs are complemented by cellular coinfection, suggesting an important role for spatial structure in viral spread. This frequent coinfection leads to higher gene copy numbers at the cellular level, consequently promoting reassortment and free mixing of genes. Thus, a reliance of IAVs on coinfection may have important implications for viral adaptation to novel environments such as new hosts following cross-species transmission.

## Methods

**Quantification of $P_P$ values**. A single-cell sorting assay was used to measure $P_P$, the probability that an individual genome segment from an infectious virion is

successfully delivered and replicated within the infected cell. The technical details are described in the "Single-cell sorting assay for measurement of $P_P$ values" section. Here, we describe the mathematical analysis used to calculate $P_P$ from the experimental data. The presence or absence of different viral genome segments was measured by qRT-PCR, with each well of a 96-well plate representing the viral RNA that was present in the cell that was initially sorted into the plate.

Given the MOI of Pan/99-WT virus used in the experiments, an appreciable number of wells are expected to receive two or more viral genomes, and so a mathematical adjustment is needed to estimate the probability of each genome segment being delivered by a single virion. Using the relationship between MOI and the fraction of cells infected from Poisson statistics, i.e., $f = 1 - e^{-MOI}$, the probability of the $i$th segment being present in a singly infected cell, or $P_{P,i}$ can be calculated from the 96-well plate using the following equation:

$$P_{P,i} = \frac{MOI_i}{MOI_{wt}} = \frac{-\ln(1 - f_i)}{-\ln(1 - f_{wt})} = \frac{\ln\left(1 - \frac{C_i}{A}\right)}{\ln\left(1 - \frac{B}{A}\right)} \quad (1)$$

where $A$ is the number of Helper$^+$ wells, $B$ is the number of WT$^+$ wells (containing any WT segment), and $C_i$ is the number of wells positive for the WT segment in question. Wells that were negative for Helper virus segments were excluded from analysis. For each experimental replicate, the geometric mean $P_P$ value was calculated to represent an average $P_P$.

**Computational simulation to predict reassortment frequency**. To predict reassortment frequency expected given the $P_P$ values measured in the single-cell sorting assay, the model described by Fonville et al.[26] was used. Briefly, this model simulates infection of a population of cells with a 1:1 mixture of WT and var viruses across a range of MOI. When a virion infects a cell, it can deliver a single copy of each segment with probability $P_P$. In this model, $P_P$ probabilities are segment-specific, but identical for WT and var viruses. A cell is deemed "HA$^+$" if it contains at least one copy of each of the PB2, PB1, PA, HA, and NP segments, and "Productive" if it contains at least one copy of each segment. The probability of a progeny virion being reassortant is calculated for each cell as

$$p(\text{Reassortant virion}) = 1 - p(\text{WT virion}) - p(\text{var virion}), \quad (2)$$

where

$$p(\text{WT virion}) = \frac{\#WT\ PB2\ copies}{\#Total\ PB2\ copies} * \frac{\#WT\ PB1\ copies}{\#Total\ PB1\ copies} * \cdots \frac{\#WT\ NS\ copies}{\#Total\ NS\ copies}, \quad (3)$$

and similar for $p(\text{var virion})$. The predicted percentage of reassortant virions resulting from a coinfection (% Reassortment) is the average of $p(\text{Reassortant virion})$ values among Productive cells. To generate the predictions shown herein, each simulation used a set of eight $P_{P,i}$ values measured in a single experimental replicate. Thus, a total of 13 separate predictions were made.

**Analysis of pairwise associations between segments**. To calculate the pairwise associations between segments, we first defined the probability that each cell acquired its combination of segments by infection with a single virion ($p(1\ \text{virion} | \text{segment combination})$), as follows. The results of this calculation are shown in Supplementary Fig. 1.

$$p(1\ \text{virion}|\text{segment combination}) = \frac{p(\text{segment combination}|1\ \text{virion}) * p(1\ \text{virion})}{p(\text{segment combination})} \quad (4)$$

The probability that one virion infected a given cell, $p(1\ \text{virion})$, was calculated using the Poisson distribution, with

$$\lambda = -\ln\left(1 - \frac{WT^+\ cells}{Helper^+\ cells}\right). \quad (5)$$

The probability of a given segment combination arising following entry of $v$ virions was calculated under the assumption of independent delivery of genome segments as

$$p(\text{segment combination}|v\ \text{virions}) = \prod_{i=0}^{8} \begin{cases} 1 - \left(1 - P_{P,i}\right)^v & \text{if segment } i \text{ present} \\ \left(1 - P_{P,i}\right)^v & \text{if segment } i \text{ absent} \end{cases}. \quad (6)$$

Using the estimated probability of segment combinations arising from infection with $v$ virions and the frequency of infection by $v$ virions, the overall probability of a cell containing each segment combination was calculated as

$$p(\text{segment combination}) = \sum_{v=0}^{\infty} p(\text{segment combination}|v\ \text{virions}) * p(v\ \text{virions}). \quad (7)$$

Because the number of cells infected and the measured $P_P$ values varied between experimental replicates, each replicate was analyzed independently to calculate $P_P$, $p(\text{segment combination} | v\ \text{virions})$, and $p(1\ \text{virion}|\text{segment combination})$ for each

cell, and all experiments were pooled for the final analysis of segment associations. Using $p(1\ \text{virion} | \text{segment combination})$ as a weighting factor for each cell, the pairwise correlations between WT segments were then calculated. Significant associations were defined as those with $p < 0.05$ after Bonferroni correction for multiple comparisons, where $Nr^2$ follows a $\chi^2$ distribution with three degrees of freedom ($N$ = sum of all $p(1\ \text{virion} | \text{segment combination})$ values).

**Probabilistic model of IVG costs for cellular infectivity**. To define the impact of incomplete viral genomes on viral infectivity, we first considered how specific infectivity, the ratio of PFUs to virus particles, changes with $P_P$. The proportion of virions that can form plaques, or the probability of productive infection resulting from a single virion, was estimated as

$$p = P_P^8 \quad (8)$$

This is a Bernoulli process with a defined probability of success or failure, so the variance was estimated as

$$\sigma^2 = p(1 - p) = P_P^8\left(1 - P_P^8\right), \quad (9)$$

and the 95% confidence interval given by $P_P^8 \pm \sigma$.

We next considered how the infectious unit (the average number of virions required to result in productive infection of a cell) varies with $P_P$. This model assumes that a single virion can deliver one copy of each segment to a cell with probability $P_P$, and that delivery of each segment is independent. The act of segment delivery by a single virion is therefore modeled as a binomial process, where $p = P_P$, $N$ = # of missing segments, and $k$ = # of segments added by one virion. For an uninfected cell, $N = 8$ missing segments, and

$$p(k\ \text{segments delivered}) = \binom{8}{k}\left(P_P^k(1 - P_P)^{8-k}\right). \quad (10)$$

For a cell that has already received some segments, each successive virion can deliver one copy of each segment not already present. We model this process of infection by successive virions as a Markov chain in which a cell can exist in nine states, containing between 0 and 8 genome segments. Transitions between states are governed by the $9 \times 9$ matrix $\mathbf{T}$, in which each element is described by the binomial distribution:

$$\mathbf{T}_{i,j} = \binom{N}{k}\left(p^k(1 - p)^{N-k}\right) = \binom{8 - i}{j - i}\left(P_P^{j-i}(1 - P_P)^{8-j}\right) \quad (11)$$

where $i$ is the number of segments a cell contains before infection, and $j$ is the number of segments it contains after infection. Since the binomial distribution is not defined for $k < 0$, all entries below the main diagonal are populated by 0 s. The state of eight segments, or productive infection, becomes the absorbing state, and it is assumed that each cell will obtain all eight genome segments given the addition of enough virions.

To estimate how many virions are required to reach the state of productive infection, we first define a $1 \times 9$ vector representing the distribution of segments per cell. Each value in this vector gives the probability that a cell contains a given number of segments. To represent an uninfected cell, we set $\tau_0 = [1,0\ldots0]$ to indicate that the probability that 0 segments are present is 1, and the probability that 1–8 segments are present is 0. The distribution of segments in a cell that has been infected with $v$ virions is then given by:

$$\tau_v = \tau_0 * \mathbf{T}^v \quad (12)$$

With the 9th element of $\tau_v$ representing the probability a cell contains eight segments, and is therefore productively infected.

Finally, we use survival analysis to calculate the expected number of virions that must infect a cell before it receives all eight segments. We first define $\mathbf{T_{sub}}$ to represent the upper-left $8 \times 8$ matrix of $\mathbf{T}$ (in which a cell contains 0–7 segments), $\tau_{sub}$ as the first eight columns of $\tau_0$, and $\mathbf{1_{sum}}$ as an $8 \times 1$ vector where each value is 1, which acts to sum each state into a single value. For a cell that starts with 0 segments, $\tau_{sub} = [1, 0\ldots0]$. The probability distribution of a cell containing 0–7 segments is given by $\tau_{sub} * I$ (where $I$ is the identity matrix) for an uninfected cell, $\tau_{sub} * \mathbf{T_{sub}}$ for one virion, $\tau_{sub} * \mathbf{T_{sub}}^2$ for two virions, and so on. For an arbitrary number of virions ($v$), the distribution is given by $\tau_{sub} * \mathbf{T_{sub}}^v$. The total probability that a cell contains 0–7 segments is then calculated as $\tau_{sub} * \mathbf{T_{sub}}^v * \mathbf{1_{sum}}$. As more virions infect a cell, this probability converges to 0. We therefore estimate the number of virions required to fully infect a cell using the equation:

$$E(v) = \tau_{sub} * \lim_{n\to\infty}(I + \mathbf{T_{sub}}^1 + \mathbf{T_{sub}}^2 + \ldots \mathbf{T_{sub}}^n) * 1_{sum} = \tau_{sub} * (I - \mathbf{T_{sub}})^{-1} * 1_{sum} \quad (13)$$

This summary statistic represents the number of transitions required for a cell to reach the absorbing state, or more simply, the average number of virions required to infect a cell. The variance on this quantity can be calculated as

$$\sigma^2 = \left(2 * (I - \mathbf{T_{sub}})^{-1} - I\right) * \tau_{sub} - \tau_{sub,sq}, \quad (14)$$

where each element of $\tau_{sub}$ is squared to generate $\tau_{sub,sq}$. The confidence interval of this estimate is then given by $E(v) \pm \sigma$. A more detailed proof of these derivations can be found in Finite Markov Chains[44].

**Table 1 Parameters used in model of spatially structured viral spread**

| Parameter | Meaning | Value | Units |
|---|---|---|---|
| $P_P$ | Probability an individual segment is replicated following infection by a single virion | Varied (0.575 or 1.0) | — |
| Burst_Rate | Virions produced per infected cell | $962/24 = 40.1$ | hour$^{-1}$ |
| Attach_Rate | Free virions attaching to local cell | $60/10 = 6$ | hour$^{-1}$ |
| Detach_Rate | Bound virions released | $10/24 = 0.417$ | hour$^{-1}$ |
| Infect_Rate | Bound virions infecting local cell | $1/6 = 0.167$ | hour$^{-1}$ |
| Death_Rate | Productively infected cells dying | $1/12 = 0.083$ | hour$^{-1}$ |
| Exclude_Rate | Infected cells becoming refractory to super-infection | $1/8 = 0.125$ | hour$^{-1}$ |
| D | Diffusion coefficient | Varied ($10^{-1}$-$10^4$) | $\mu m^2\ s^{-1}$ |

**Probabilistic model of IVG costs for population infectivity**. To define the impact of incomplete viral genomes on the ability of a virus population to establish infection in a population of cells, the probabilistic model described above was adapted to account for the Poisson distribution of virions among a population of cells. It is assumed that one productively infected cell produces enough virions to infect other cells in subsequent rounds of replication, and so establishing infection in a population of cells requires that at least one cell receive all 8 genome segments. For a given MOI, the probability of a cell being infected by $v$ virions follows the Poisson distribution

$$p(v) = \frac{MOI^v e^{-MOI}}{v!}. \qquad (15)$$

At each $v$, the probability that a cell received any given segment is equal to $1 - (1 - P_P)^v$, and so the probability that a cell is productively infected after infection with $v$ virions is

$$p(8|v) = (1 - (1 - P_P)^v)^8. \qquad (16)$$

The sum of the joint probabilities $p(v) * p(8|v)$ across all values of $v$ gives the probability that any given cell is productively infected:

$$\lim_{N\to\infty} \sum_{v=1}^{N} p(v) * p(8|v). \qquad (17)$$

Multiplying this probability by the number of cells in the population gives the expected number of cells infected, and the probability of the population becoming infected is equal to this value or 1, whichever is lower. The ID$_{50}$ was estimated as the lowest MOI yielding a probability $\geq 50\%$. A similar analysis was used to estimate the ID$_{50}$ when complementation was not allowed, with the $p(8|v)$ function being modified to

$$p(8|v) = 1 - \left(1 - P_P^8\right)^v \qquad (18)$$

to reflect the fact that only complete viral genomes could initiate infection. Finally, the percentage of infected cells that contained incomplete viral genomes was calculated by estimating the probability that a cell infected by $v$ virions contained between one and seven segments,

$$p(1-7|v) = 1 - p(8|v) - p(0|v) = 1 - \left(1 - P_P^8\right)^v - (1 - P_P)^{8v} \qquad (19)$$

and determining the total proportion of infected cells containing IVGs using the equation

$$\% \text{ Cells with IVGs} = \lim_{N\to\infty} \sum_{v=1}^{N} p(v)^* \frac{p(1-7|v)}{p(1-7|v) + p(8|v)} * 100. \qquad (20)$$

**Individual-based model of replication**. A cellular automaton model of viral spread was developed to investigate the relationships between spatial structure, prevalence of incomplete viral genomes, and viral fitness. The system consists of a $100 \times 100$ grid of cells. Each cell contains 0–8 distinct IAV genome segments, and additional copies of the same segment are assumed to be redundant. Virions exist on the same grid, in a bound or unbound state. When a virion infects a cell, any missing segments may be delivered, with the probability of delivery defined by $P_P$, as derived in Figs. 1 and 2.

The simulation begins with a single productively infected cell in the middle of the grid. The following events occur at each time-step (3 min), and the frequency of each of these events is governed by the parameters listed in Table 1.

1. All virions not currently bound to a cell will diffuse. First, the total distance traveled is randomly drawn from the normal distribution

$$\text{Distance}_{\text{Total}} = N(\mu = 0, \sigma = \sqrt{2Dt}) \qquad (21)$$

where $D$ is the diffusion coefficient, $t$ is the length of the time-step, $\mu$ is the mean, and $\sigma$ is the standard deviation. Second, the direction traveled is randomly drawn from the uniform distribution

$$\theta = U(0, 2\pi). \qquad (22)$$

The total distance traveled is then converted to orthogonal distances

$$\text{Distance}_X = \cos(\theta)^* \text{Distance}_{\text{Total}} \qquad (23)$$

and

$$\text{Distance}_Y = \sin(\theta)^* \text{Distance}_{\text{Total}} \qquad (24)$$

Distance and direction of travel are calculated independently for each virion. These distances are then used to adjust the $X$ and $Y$ positions of the diffusing virions, which are tracked in absolute units ($\mu$m). Cells are modeled as squares measuring $30\,\mu m \times 30\,\mu m$, and so a virion with an absolute position $0 \leq X < 30$ corresponds to cell position $X = 1$, one with absolute position $30 \leq X < 60$ corresponds to cell position $X = 2$, and so on. If a virion would diffuse beyond the border of the grid, it instead emerges from the other side (e.g., a virion that would be moved to the cell position $[X = 105, Y = 69]$ is instead placed at the position $[X = 5, Y = 69]$).

2. Free virions may attach to the cell at their current position. While attached, virions are unable to diffuse.

3. Bound virions may be released, or infect the cell to which they are attached. Virions that are released become free-floating, and are able to diffuse. When virions infect a cell, the number of segments added to that cell is determined by the probabilistic model described above. The number of segments added is calculated from the binomial distribution

$$B(N = 8 - S, p = 1 - (1 - P_P)^v) \qquad (25)$$

where $S$ is the number of segments the cell already contains, $P_P$ is the probability an individual segment is delivered successfully, and $v$ is the number of virions infecting the cell at the current time-step. For example, if a cell already contains five segments, additional virions infecting it may add 0–3 segments ($N = 8 - 5 = 3$). If four virions are infecting this cell, then the probability of a given segment being delivered is high ($p = 1 - (1-0.58)^4 = 0.97$), and the probability of four virions all failing to deliver a given segment $((1-0.58)^4 = 0.03)$ is relatively low.

4. Infected cells (those containing 1–8 segments) may become refractory to superinfection. Diffusing virions cannot bind to nonsusceptible cells, and any currently bound virions that attempt to infect these cells automatically fail to deliver all segments.

5. Productively infected cells (containing eight segments) produce virions, which are initially bound to the producer cell's surface. The number of virions produced by each cell is independently drawn from the Poisson distribution with $\lambda = \text{Burst\_Rate}$ (962 virions day$^{-1}$ or 2.06 virions time-step$^{-1}$ (time-step = 3 min)). Each virion is placed at a random location on the producer cell as determined by the uniform distribution.

6. Productively infected cells may die. These cells lose all segments, cannot produce virions, and cannot be bound by diffusing virions.

At each time-step, the number of virions that bind, release, or infect, and the number of cells that die or become refractory to superinfection, are calculated using the Poisson distribution with $\lambda = \text{Rate} * N$, where rates are described in Table 1, and $N$ represents the number of virions or infected cells present at that time.

To generate the data shown in Figs. 4 and 6b, these events were iterated over multiple rounds of infection up to 96 h post infection. Ten simulations per ($D$, $P_P$) combination were conducted.

**Cells**. Madin–Darby canine kidney (MDCK) cells (contributed by Peter Palese, Icahn School of Medicine at Mount Sinai) were cultured in the minimal essential medium (MEM) supplemented with 10% fetal calf serum (FCS), penicillin (100 IU), and streptomycin (100 µg/mL). 293T cells (ATCC, CRL-3216) were cultured in the Dulbecco's modified essential medium (DMEM) supplemented with 10% FCS.

As used herein, "complete medium" refers to MEM supplemented with 10% FCS and penicillin/streptomycin at the above concentrations, which was used for

maintaining cells in culture. Following infection with influenza viruses, cells were incubated with "virus medium", which herein refers to MEM supplemented with 0.3% bovine serum albumin and penicillin/streptomycin at said concentrations. When their presence is indicated, TPCK-treated trypsin was used at 1 μg/mL, $NH_4Cl$ at 20 mM, HEPES at 50 mM.

**Viruses.** All viruses were generated by reverse genetics following modification of the influenza A/Panama/2007/99 (H3N2) virus cDNA, which was cloned into pDP2002[45]. All viruses were cultured in 9–11-day old embryonated hens' eggs unless otherwise noted below. To limit propagation of defective interfering viral genomes, virus stocks were generated either from a plaque isolate or directly from 293T cells transfected with reverse genetics plasmids. Defective interfering particle contents were quantified using a digital droplet PCR (ddPCR) assay and confirmed to be minimal[46]. The only genetic modification made to the Pan/99-WT virus was the addition of sequence encoding a 6-His tag plus GGGGS linker following the signal peptide of the HA protein[8]. A genetically distinct but phenotypically similar virus, referred to herein as "Pan/99-Helper", was generated by the introduction of six or seven silent mutations on each segment, as well as the addition of the HA-tag (sequence: YPYDVPDYA) instead of the 6-His tag. The silent mutations are listed in Supplementary Table 1, and were designed to introduce strain-specific primer-binding sites, allowing the presence or absence of each segment to be measured by qRT-PCR. Epitope tags in HA allowed identification of infected cells by flow cytometry.

To visualize infected cells by microscopy, a virus which expresses a tetracysteine (TC) tag on the NP protein, referred to herein as "Pan/99-NP_TC", was generated by the introduction of the amino acid sequence CCPGCC at the C-terminus of NP. To avoid disruption of the packaging of the NP segment, 150 nt corresponding to the 3′ end of the NP ORF was duplicated following the TC tag sequence and the stop codon.

A virus with two distinct forms of the M segment, referred to herein as "Pan/99-M.STOP virus", was constructed. Site-directed mutagenesis was used to introduce nonsense mutations into the pDP2002 plasmid containing the sequence of the M segment in order to abrogate expression of M2. but not M1 (M1.Only), or vice versa (M2.Only). For M2.Only, three in-frame stop codons were introduced downstream of the sequence encoding the shared M1/M2 N-terminus (the nucleotide changes introduced were T55A, C75G, and A86T). An in-frame ATG at nucleotide 152 was also disrupted by mutation to TTG. For M1.Only, three in-frame stop codons were introduced in the M2 coding region downstream of the M1 ORF (T820A, T826A, and CC786, 787TA). In addition, four nonsynonymous changes were made to M2 coding sequence in the region following the splice acceptor site and upstream of the introduced stop codons. These changes were synonymous in the M1 reading frame. Both plasmids were used in conjunction with pDP plasmids encoding the other seven segments to generate a mixed virus population, in which each virion contained an M1.Only or M2.Only segment. At 24 h post transfection, 293T cells were washed with 1 mL of PBS, then overlaid with $1 \times 10^6$ MDCK cells in virus medium plus TPCK-treated trypsin, and incubated at 33 °C for 48 h. Supernatant was used to inoculate a plaque assay, and after 48 h a plaque isolate was used to inoculate a 75 cm² flask of MDCK cells. Following 48 h of growth, this stock was aliquoted and used to inoculate a plaque assay. One plaque isolate was diluted and used to inoculate 10-day-old embryonated chickens' eggs for a third passage. Experiments were conducted with this egg passage stock.

**Infections.** Six-well dishes (Corning) were seeded with $4 \times 10^5$ MDCK cells in 2 mL of complete medium, then incubated for 24 h. Prior to inoculation, complete medium was removed, and cells were washed twice with 1 mL of PBS per wash. Inocula containing virus in 200 μL PBS were added to cells, which were incubated on ice (to permit attachment but not viral entry) for 45 min. After attachment, the monolayer was washed with PBS to remove unbound virus before 2 mL of virus medium was added and plates were incubated at 33 °C. For multi-cycle replication, TPCK-treated trypsin was added to virus medium to a final concentration of 1 μg/mL. When single-cycle conditions were required, virus medium was removed after 3 h and replaced with 2 mL of virus medium containing $NH_4Cl$ and HEPES.

**Single-cell sorting assay for measurement of $P_P$ values.** In all, $4*10^5$ MDCK cells were seeded into a six-well dish, then counted the next day just before inoculation. Cells were then washed three times with PBS and co-inoculated with the virus of interest (Pan/99-WT, MOI = 0.5 PFU/cell) and helper virus (Pan/99-Helper, MOI = 3.0 PFU/cell) in a volume of 200 μL. Cells were incubated at 33 °C for 60 min, after which they were washed three times with PBS, and 2 mL of virus medium was added. After incubation at 33 °C for 60 min, medium was removed, and cells were washed three with PBS before addition of Cell Dissociation Buffer (Corning) containing 0.1% EDTA (w/v) to release cells from the plate surface. Cells were harvested by resuspension in complete medium, followed by a series of three washes in 2 mL of FACS buffer (2% FCS in PBS). Cells were resuspended in PBS containing 1% FCS, 10 mM HEPES, and 0.1% EDTA, and filtered immediately prior to sorting on a BD Aria II. After gating to exclude debris and doublets, one event was sorted into each well of a 96-well plate containing MDCK cell monolayers at 30% confluency in 50 μL of virus medium containing TPCK-treated trypsin. After sorting, an additional 50 μL of medium was added to a final volume of 100 μL per well, and plates were spun at 1800 rpm for 2 min to help each sorted

cell attach to the plate surface. Plates were incubated at 33 °C for 48 h to allow outgrowth of virus from this single infected cell.

RNA was extracted from infected cells using a ZR-96 Viral RNA Kit (Zymo Research) as per the manufacturer's instructions. The extracted RNA was converted to cDNA using universal influenza primers (shown in Supplementary Table 2), Maxima RT (Thermo Scientific, 100 U/sample) and RiboLock RNase inhibitor (Thermo Scientific, 28 U/sample) according to the manufacturer's instructions. After conversion, cDNA was diluted 1:4 with nuclease-free water and used as a template (4 μL/reaction) for segment-specific qPCR using SsoFast EvaGreen Supermix (Bio-Rad) in 10 μL reactions. Primers for each segment of Pan/99-WT virus, as well as the PB2 and PB1 segments of Pan/99-Helper virus, are given in Supplementary Table 2, and were used at final concentrations of 200 nM each.

**Flow cytometry.** At 12 h post inoculation, virus medium was aspirated from infected cells, and monolayers were washed with PBS. The monolayer was disrupted using 0.05% trypsin + 0.53 mM EDTA in Hank's balanced salt solution (HBSS). After 15 min at 37 °C, plates were washed with 1 mL of FACS buffer (PBS + 1% FCS + 5 mM EDTA) to collect cells, and transfered them to 1.7-mL tubes. Cells were spun at 2500 rpm for 5 min, then resuspended in 200 μL FACS buffer and transferred to 96-well V-bottom plates (Corning). The plate was spun at 2500 rpm and supernatant discarded. Cells were resuspended in 50 μL of FACS buffer containing antibodies at the following concentrations, then incubated at 4 °C for 30 min:

(1) His Tag-Alexa 647 (5 μg/mL) (Qiagen, catalog no. 35370)
(2) HA-Tag-FITC (7 μg/mL) (Sigma, clone HA-7, catalog no. H7411)

After staining, cells were washed by three times by centrifugation and resuspension in FACS buffer. After the final wash, cells were resuspended in 200 μL of FACS buffer containing 7-AAD (12.5 μg/mL) and analyzed by flow cytometry using a BD Fortessa.

This approach was modified slightly when staining for M1 and M2. After staining for His and HA (where indicated), cells were washed once with 200 μL of FACS buffer, then resuspended in 100 μL of BD Cytofix/Cytoperm buffer and incubated at 4 °C for 20 min. BD Cytoperm/Cytowash (perm/wash) buffer was added to each well, and cells were spun at 2500 rpm for 5 min. After a second wash, cells were resuspended in 50 μL perm/wash buffer containing antibodies at the following concentrations:

(1) Anti-M1 GA2B conjugated to Pacific Blue (4 μg/mL) (ThermoFisher, catalog no. MA1-80736 (antibody) and P30013 (conjugation kit))
(2) Anti-M2 14C2 conjugated to PE (4 μg/mL) (Santa Cruz, catalog no. sc-32238 PE)

Following another 30 min of staining at 4 °C, cells were washed three times (as described above) with perm/wash buffer, then resuspended in FACS buffer without 7-AAD just prior to analysis on the BD Fortessa.

**Single-cycle growth curves.** Cells were inoculated with Pan/99-WT virus at MOIs of 1, 3, 6, 10, or 20 PFU/cell, and incubated with 2 mL of virus medium at 33 °C. After 3 h, virus medium was replaced with virus medium containing $NH_4Cl$ and HEPES. In all, 100 μL of medium was collected at 3, 4.5, 6, 8, 12, 18, 24, and 48 h post inoculation (with replacement by fresh medium to keep volumes consistent) for virus quantification by plaque assay. At 12 h post inoculation, cells were harvested and stained for analysis of HA expression by flow cytometry.

**Impact of secondary spread on complementation of incomplete genomes.** To optimize the approach of using Pan/99-Helper to activate and thereby detect cells containing incomplete genomes, we co-inoculated cells with a low MOI (0.01 PFU/cell) of Pan/99-WT virus and a range of Pan/99-Helper virus MOIs and measured expression of WT HA after 12 h. We observed a biphasic relationship between helper virus MOI and the benefit provided to WT virus (Supplementary Fig. 4C). As more Helper virus was added, the percentage of cells expressing WT HA initially increased as more cells became coinfected and thus capable of expressing the WT HA protein. But, as the Helper MOI increased further, a competitive effect was observed and the probability of detecting WT HA expression was decreased. Observing that Pan/99-Helper virus provided the greatest benefit—a twofold increase in the frequency of WT HA expression—at an MOI of 0.3 PFU/cell, we used that amount in further complementation experiments. Based on measured $P_P$ values, this dose is estimated to contain an average of 27 particles/cell.

In single-cycle replication conditions, cells were inoculated on ice with Pan/99-WT virus over a range of MOIs (0.1, 0.3, 0.6, 1 PFU/cell) and, at the same time, with Pan/99-Helper virus (MOI = 0.3 PFU/cell) or PBS. After inoculation, cells were washed with 1 mL of PBS, 2 mL of virus medium (no trypsin) was added, and cells were incubated at 33 °C for 3 h, after which initial virus medium was replaced with virus medium containing $NH_4Cl$ and HEPES. At 12 h post inoculation, cells were collected and stained for WT and Helper HA expression as described above.

In multi-cycle replication conditions, cells were inoculated on ice with Pan/99-WT virus at an MOI of 0.01 or 0.002 PFU/cell, and then incubated at 33 °C with virus medium containing TPCK-treated trypsin to allow for multi-cycle growth. After 12 h, cells were washed with 1 mL of PBS, then inoculated on ice with Pan/99-Helper virus (MOI = 0.3 PFU/cell), or PBS. After inoculation, cells were washed

with 1 mL of PBS, 2 mL of virus medium (no trypsin) was added, and cells were incubated at 33 °C for 3 h, after which initial virus medium was replaced with virus medium containing $NH_4Cl$ and HEPES. At 12 h post inoculation with Pan/99-Helper virus, cells were collected and stained for WT and Helper HA expression as described above. The amount of complementation provided by Pan/99-Helper virus was calculated using the equation:

$$\%\text{Enrichment} = \frac{\%WT^+|Helper^+ - \%WT^+|Helper^-}{\%WT^+|Helper^-} \times 100 \quad (24)$$

where %WT⁺ | Helper⁺ denotes the percentage of cells expressing WT HA in the Helper HA⁺ sub-population.

**Microscopy.** To visualize foci of infection, cells were infected with Pan/99-NP_TC virus at an MOI of 0.002 PFU/cell, and incubated under multi-cycle conditions. After 12 h, media was aspirated and changed to virus medium containing $NH_4Cl$ + HEPES, and cells were incubated for another 12 h. To visualize randomly dispersed infections, cells were inoculated with Pan/99-NP_TC virus at an MOI of 0.1 PFU/cell, and incubated under single-cycle conditions for 12 h (3 h with virus medium, followed by 9 h with virus medium containing $NH_4$ Cl + HEPES). To visualize FlaSH reagent background, cells were mock-infected with 200 μL of PBS, then incubated under single-cycle conditions for 12 h (3 h with virus medium, followed by 9 h with virus medium containing $NH_4$ Cl + HEPES).

At the end of the infection, cells were washed twice with 1 mL of PBS, then stained with 2 μM FlaSH reagent (ThermoFisher, cat. no. T34561) diluted in Opti-MEM. During staining, plates were incubated at 37 °C for 30 min. To remove FlaSH, cells were washed three times by adding 300 μL of BAL wash buffer diluted to 250 μM in Opti-MEM and incubating at 37 °C for 10 min per wash. After washing, cells were washed once with 1 mL of PBS, then fixed by addition of 500 μL 4% paraformaldehyde and incubation for 10 min at room temperature. After fixation, cells were stained with 300 nM DAPI for 3 min at room temperature, then washed three times with 1 mL of PBS, and visualized using a BioTek Lionheart FX. For each image, brightness was enhanced by 20%, and contrast by 40%.

**Digital droplet PCR (ddPCR).** Primers and probes (listed in Supplementary Table 3) were diluted to concentrations of 900 nM and 250 nM per primer and probe, respectively. Overall, 22 μL of reactions were prepared with 11 μL of Bio-Rad SuperMix for Probes (1× final concentration), 6.6 μL of diluted primers (900 nM/primer, final concentration) and probes (250 nM/probe, final concentration), and 4.4 μL of diluted cDNA. In all, 20 μL of each reaction mixture was partitioned into droplets using a Bio-Rad QX200 droplet generator per the manufacturer's instructions. PCR conditions were: (1) 95 °C for 10 min, (2) 40 cycles of A) 94 °C for 30 s and B) 57 °C for 1 min, (3) 98 °C for 10 min, and hold at 4 °C. Droplets were then read on Bio-Rad QX200 droplet reader, and the number of cDNA copies/μL was calculated.

**Guinea pig infections.** Female Hartley guinea pigs were obtained from Charles River Laboratories (Wilmington, MA) and housed by Emory University Department of Animal Resources. All experiments were conducted in accordance with the Guide for the Care and Use of Laboratory Animals of the National Institutes of Health. The protocol was approved by the Emory University Institutional Animal Care and Use Committee, and in accordance with established guidelines and policies at Emory University School of Medicine (Protocol Number: DAR-2002738-ELMNTS-A). For $ID_{50}$ estimation and analysis of viral shedding, guinea pigs were anesthetized by intramuscular injection with 30 mg/kg ketamine/ 4 mg/kg xylazine, then inoculated intranasally with 300 μL of virus diluted in PBS. Nasal washes were collected in 1 mL of PBS on days 1, 2, 3, 5, and 7 by anesthetizing guinea pigs, placing them flat on an elevated surface, and gently rinsing the nasal cavities so that PBS dripped onto a sterile Petri dish[47]. Virus present in nasal washes was then titered by RT-ddPCR targeting the NS segment. For transmission experiments, inoculated guinea pigs were individually housed in Caron 6040 environmental chambers at 10 °C and 20% relative humidity. At 24 h post inoculation, one naive guinea pig was introduced to each cage with one inoculated animal. Nasal washes were collected on days 2, 4, 6, and 8, and titered by RT-ddPCR.

**Reporting summary.** Further information on research design is available in the Nature Research Reporting Summary linked to this article.

## Data availability

All raw data supporting these findings and used to generate figures and Supplementary Information are available on Github at the following URL: https://github.com/njacobs627/Pan99_IVGs_Spatial_Structure. The data underlying Figs 1a–c, 2a–d, 3a, b, 4a–d, 5a–c, 6a–c, 7b–d, 8a–d and Supplementary Figs. 1, 2a–d, 3a–c, 4c–d and 5b are provided as a source data file.

## Code availability

All R code used for simulations, data analysis, and visualizations is available on Github at the following URL: https://github.com/njacobs627/Pan99_IVGs_Spatial_Structure.

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

## Acknowledgements

We thank Daniel Perez for the pDP2002 plasmid and Shamika Danzy and Hui Tao for technical assistance. This work was funded in part by the NIH/NIAID Centers of Excellence in Influenza Research and Surveillance (CEIRS), contract number HHSN272201400004C (to A.C.L. and J.S.), and by NIH/NIAID grants R01 AI099000 (to ACL) and U19 AI117891 (to R.A., A.C.L., and J.S.).

## Author contributions

N.T.J. contributed to the conception of work, experimental design, data acquisition and analysis, interpretation of data, and writing of the paper; N.O.O. contributed to data acquisition and analysis; A.A. contributed to data analysis and interpretation; J.S. contributed to experimental design, interpretation of data, and writing of the paper; R.A. contributed to conception of work, interpretation of data, and writing of the paper; A.C.L. contributed to conception of work, experimental design, interpretation of data, and writing of the paper.

## Additional information

**Competing interests:** The authors declare no competing interests.

