## [Peer Review File · Nature Communications]

Reviewers' Comments:

Reviewer #1:

Remarks to the Author:

This paper by Jacobs et al seeks to quantify the abundance of influenza virions that lack the full viral genome and to understand the role of co-infection in supporting the replication of these incomplete viral genomes (IVGs). They use a clever, helper virus-based system to enumerate the fraction of virions that are incapable of replicating each of the eight genome segments, and find that this number is surprisingly high: on average only 58% of virions express a given genome segment, and only 1.3% of virions have the full genome. They combine *in silico* modeling with well-designed cell culture experiments to both precisely quantify the costs of IVGs in terms of the increased number of virions needed to initiate productive infection, and to suggest how spatial structuring of infection and high MOI may mitigate these costs. They go on to test the potential for high MOI infection to rescue IVGs *in vivo* by infecting guinea pigs with a virus population that depends upon complementation to replicate. Surprisingly, they show that this complementation-dependent virus replicates to relatively high titers in the animal, but is unable to transmit. Overall, the results reveal important yet complicated roles for IVGs, virion diffusion, and MOI during influenza infection.

The manuscript is very well written, the approaches are novel, and the conclusions are largely supported by the data. The pairing of modeling work with experimentation is especially powerful. Overall, while a few experiments are needed to rule out alternative explanations for some of the key experiments, this is very compelling work that provides a significant conceptual advance in our understanding of influenza virus replication dynamics and will be of wide interest.

Major comments:

- In Figure 6C, the decreased complementation observed when helper virus is added 12 hours post WT infection versus simultaneously is used to support the idea that limited diffusion during secondary spread promoted complementation of WT particles. An alternative explanation is that superinfection immunity (shown in Sun et al (10.1128/mBio.01761-18) and Dou et al (10.1016/j.celrep.2017.06.021)) triggered by the WT virus limits the ability of helper virus to complement when it is added 12 hours later. The authors need to distinguish between these two possibilities. Also, in line 243 and elsewhere: The authors indicate that virus produced during secondary replication in cell culture will be locally dispersed, resulting in spatial structuring. Is there any experimental data to support the extent to which this is true?

- The M.STOP experiments are very clever, and the guinea pig experiments are especially intriguing. My concern here is whether the virus is really absolutely dependent upon complementation as the authors describe it. The dilution series in Figure 7C never goes low enough to demonstrate the absence of infectious units that co-express M1 and M2. It is possible that some degree of virion aggregation in the stock means that fully replication competent infectious units are always present at some level. The lack of absolute requirement for M2 also complicates things, as the authors point out, meaning that M2- virions may just be attenuated rather than dead in the absence of complementation. It seems possible that *in vivo* growth could entirely be driven by the M2-dead component of the M.STOP population without any complementation, making it hard to interpret the *in vivo* results. This could be clarified by a couple of experiments. First, how does an M2-dead mutant (in which all virions express M1 and none express M2) compare with the M.STOP virus? Second, if guinea pigs are infected with either WT virus at both the dose used in Figure 8A or with an 815-fold reduction to reflect the difference in infectivity seen for M.STOP in Figure 7D, would the comparative viral load data look like that in Figure 8A? In other words, to what extent are the growth curves in Figure 8A influenced by the fact that the inoculum doses for the two viruses may have differed by roughly 800-fold in terms of guinea pig ID50?

Minor comments:

- Line 62: Should cite Russell et al (10.7554/eLife.32303) which showed heterogeneity in influenza virus gene expression here

- Line 232: Referring to cells as "semi-infected" sounds awkward. Maybe better to say the proportion of infected cells that lack the complete viral genome.

Reviewer #2:

Remarks to the Author:

In Jacobs et al, the authors perform a series of elegant experimental and modeling studies to examine the complex feature of influenza viral RNA packaging efficiency, spread and reassortment. The study sheds light on many important areas of influenza biology that are hotly debated. Using a genetically similar viruses, the authors examine the 'productive presence' (termed Pp) of individual gene segments in a single cell level. They found that the Pp was much lower than would be expected for viruses that efficiently package all eight copies of a viral RNA. Based on sophisticated modeling they calculate that only $\sim 1.3\%$ of a virion progeny will produce an infection where all 8 segments are replicated and present. Therefore, co-infection of cells with multiple incomplete viral genomes is necessary for replication fitness within hosts but is dependent upon the spatial diffusion of viruses within respiratory tracts and burst size. This is evident in the creative use of a novel virus that requires co-infection to replicate, the M.stop virus will only replicate when a single cell is infected with multiple viruses encoding either M1.stop or M2.stop. Inoculation of guinea pig with the M.stop viruses demonstrates that complementation of viruses with incomplete genome is capable in an experimental infection scenario, since robust viral replication in guinea pig upper respiratory tract was observed. However, these viruses are incapable of overcoming the bottleneck in transmission to recipient animals, likely because so few virions initiate a natural infection.

These studies are very intriguing and call into question the prevailing dogma that the majority of virions contain all 8 segments. Based on the data presented in this manuscript using the single cell assay, it seems there may be a continuum, where progeny virions could contain less than all 8 segments and/or upon cellular entry not all viral RNA are transported into the nucleus with equal affinity. The authors do a nice job in the discussion detailing all possible explanations for their observations and how it fits into previous work on semi-infectious particles. A few specific comments are detailed below that might enhance the story.

Specific comments:

1. The single cell data elucidating the modest Pp for all 8 segments (Figure 1) is very compelling. This reviewer has three suggestions that may be useful points of discussion regarding the single cell experiments:

- a. As the authors mention in the discussion, the Pp could change based on strain type. Could the authors either report the Pp for other biological strains, or would it be possible to model the reassortment potential between strains with differing Pp phenotype? Along this same line would it be possible to attribute varying Pp values to specific segments and assess whether this would alter the relationship between segments?
- b. Does the Pp of a viral stock change over time? One would assume, that a low MOI infection could produce more incomplete viral genomes since only $\sim 1.3\%$ would contain all 8 segments from a given single infection.
- c. In lieu of a helper virus, would it be feasible to perform a similar analysis on cell lines expressing the complementing polymerase complex? This would remove any bias that may be present with a co-

viral infection. Presumably a similar Pp would be generated in this case as well.

2. While the majority of the manuscript was very well written and clear, further explanation of some of the modeling aspects would be helpful, especially in describing figure 4 parts B and C with the diffusion constants.

Minor comments:

1. Line 15 – the second 'the' should read then.
2. Please state the timepoint used to generate Figure 5B.

Reviewer #3:

Remarks to the Author:

The authors follow-up on the prior work of their own group (Fonville et al, PLoS Path 2015, ref [24]) and others (Heldt et al, Nat Comm 2015, ref [20]; Brooke et al, JVI 2013, ref [19]) demonstrating the existence and effects of so-called 'incomplete (influenza) viral genomes' (IVGs) with substantial new experimental and theoretical results. The major results are:

* The development of a robust, simple, and novel experimental assay to detect the frequency at which each of the eight influenza genome segments is ultimately available for a cell's infection when that infection is accomplished with a single virion. Using this assay, the probability of occurrence of each segment is constrained to a narrow range (0.5-0.6), and a simple calculation leads to an estimate of the fraction of singly-infected cells that have a complete set (1.3 percent). [Figure 1A]

* The experimentally-measured frequency of segments are then used to parameterize a stochastic simulation (introduced in [24]), which in turn is able to fully predict the experimentally-measured frequency of reassortment in influenza virions (also from [24]) . As they describe in the manuscript (line 125), this "indicates that IVGs fully account for the levels of reassortment observed"

* A relatively simple probability model for infection, parameterized by the above-measured probability of occurrence of segments, is used to predict the expected number of viral entries to yield a productive infection (3.7 entry events).

* An agent-based computational model is used to demonstrate that constricted dispersal of newly-produced virions by a cell (mediated by a finite diffusion constant in a liquid medium) can allow for high-MOI infections of nearby cells and thus allow for the regular presence of a full complement of genomic segments. This effect was found to be consistent with their in vitro experiments.

* By developing a mutant virus that required reassortment for propagation (i.e., while wild-type virions are very unlikely to contain the full complement of genomic segments, none of the constructed mutants did), they demonstrated that a relatively normal (though attenuated) infection could be established. But, they also show that transmission, which likely occurs through "low-MOI" infections by single virions, was blocked for this mutant strain, indicating that "rare complete IAV genomes may be critical for transmission to new hosts".

Each of these accomplishments is important to the field, novel, and will spur future research. The combined impact of all of them makes this a significant work. Therefore, I am very supportive of publication of this manuscript in Nature Communications. Nevertheless, I have some suggestions for the authors:

- 1) In a few places, some disambiguation of the theoretical methods would be appreciated and useful

to the reader. The equation used to calculate Pp (probability present of the ith segment) from the FACS results, which appears after line 624, could be explained with a few more lines. For example, "Using the relationship between MOI and the fraction of cells infected from Poisson statistics, i.e., $f = 1 - \exp(-\text{MOI})$, the probability of the ith segment can be calculated from the 96-well infections as:

$$P_{pi} = \text{MOI}_i / \text{MOI}_{wt} = -\ln(1 - f_i) / -\ln(1 - f_{wt}) = \ln(1 - C/A) / \ln(1 - B/A)"$$

Also, it is repeatedly stated in the text that this is a "correction factor" when in fact it is just the equation from which one expects to calculate the correct value of Ppi (the equation is not a multiplicative "factor").

Within the Methods section on the probability model, it would be helpful to the reader to briefly show that the basic definition of the expectation $E[v]$ (weighted average of the v values) leads to the equation after line 478, via " $[I - T]^{-1} = I + T + T^2 + \dots$ ".

Finally, within the Methods the description of the "Monte Carlo simulations" should precede the probability model, since its use does in the Results, and perhaps it and the probability model should be given their own short sub-sections, as is given to the agent-based model.

2) Some more details about the agent-based model would be useful, since there are some decisions that must have been made in this simulation that are not completely explained. For example, how are the rates from Table 1 implemented in the dynamics, how is the diffusion implemented (e.g., is a random distance and a random direction vector chosen?), do the agent based dynamics match those of an ODE model (for some test cases) in the limit of high diffusion? These could be addressed in a section of the Supplemental Material, rather than expanding the Methods.

3) Uncertainties and/or confidence intervals could be associated to the values obtained from the various experiments and models and added to the text. For example the 0.58 value for Pp, the 3.7 virions value for the "infectious unit", the 11.5 PFU burst size.

4) The Introduction provides a coherent and complete introduction to the problem at hand. This reader would have liked to hear a little bit more about the issue discussed in lines 67-71 --- i.e., the origin of IVGs, if most isolated virions are found to be complete --- either within the Introduction or in the Discussion. It is determined (and then taken for granted) throughout the manuscript that singly-infected cells have an incomplete set of viable genome segments, yet the evidence presented in those lines seems to say that virions entering cells are completed. Some speculation or discussion in the literature would be valuable here. (Although perhaps no more can be said than the statement in the Discussion on line 369.)

5) The limited burst size of single-cycle infections is common for high-MOI ("single-cycle") infections, and is usually associated to the effects of defective interfering particles (see, e.g., Liao et al, Royal Society Interface 13: 20160412 (2016)). This should probably be commented-on here.

6) I found a few typos (e.g., lines 24 and 28 in the abstract, line 475; the variable "k" is not defined before line 468; what are the meaning of the commas/slashes in the "Values" table), so the whole text should be checked again.

n

Reviewer #1

This paper by Jacobs et al seeks to quantify the abundance of influenza virions that lack the full viral genome and to understand the role of co-infection in supporting the replication of these incomplete viral genomes (IVGs). They use a clever, helper virus-based system to enumerate the fraction of virions that are incapable of replicating each of the eight genome segments, and find that this number is surprisingly high: on average only 58% of virions express a given genome segment, and only 1.3% of virions have the full genome. They combine in silico modeling with well-designed cell culture experiments to both precisely quantify the costs of IVGs in terms of the increased number of virions needed to initiate productive infection, and to suggest how spatial structuring of infection and high MOI may mitigate these costs. They go on to test the potential for high MOI infection to rescue IVGs in vivo by infecting guinea pigs with a virus population that depends upon complementation to replicate. Surprisingly, they show that this complementation-dependent virus replicates to relatively high titers in the animal, but is unable to transmit. Overall, the results reveal important yet complicated roles for IVGs, virion diffusion, and MOI during influenza infection.

The manuscript is very well written, the approaches are novel, and the conclusions are largely supported by the data. The pairing of modeling work with experimentation is especially powerful. Overall, while a few experiments are needed to rule out alternative explanations for some of the key experiments, this is very compelling work that provides a significant conceptual advance in our understanding of influenza virus replication dynamics and will be of wide interest.

We thank the reviewer for these positive comments.

In Figure 6C, the decreased complementation observed when helper virus is added 12 hours post WT infection versus simultaneously is used to support the idea that limited diffusion during secondary spread promoted complementation of WT particles. An alternative explanation is that superinfection immunity (shown in Sun et al (10.1128/mBio.01761-18) and Dou et al (10.1016/j.celrep.2017.06.021)) triggered by the WT virus limits the ability of helper virus to complement when it is added 12 hours later. The authors need to distinguish between these two possibilities.

We agree that super-infection exclusion could interfere with complementation by Helper virus, and performed a new analysis of the data to determine whether there was evidence of superinfection exclusion in the previous experiments. These results are discussed within the text at Lines 293 – 303, as follows:

“To account for the alternative possibility that spread of the WT virus over 12 h would reduce the potential for complementation by super-infection exclusion, we analyzed the level of Helper virus infection in relation to the presence of WT virus (Supplementary Figure 4D). We observed that a similar fraction of cells expressed Helper HA in i) infections with Helper virus alone (data plotted at 0% WT HA+), ii) simultaneous co-infections of WT and Helper viruses and iii) infections where Helper virus was added 12 h after low MOI WT virus. If super-infection exclusion was indeed limiting the ability of Pan/99-Helper virus to complement IVGs among cells, we would expect lower frequencies of Helper HA expression in populations where Pan/99-Helper virus was added 12 h after WT virus. Although super-infection exclusion develops in less than 12 h⁸, this outcome was expected because, under the multicycle conditions used, the vast majority of infected cells present at 12 h were recently infected in a second or third round of multiplication.”

In line 243 and elsewhere: The authors indicate that virus produced during secondary replication in cell culture will be locally dispersed, resulting in spatial structuring. Is there any experimental data to support the extent to which this is true?

To address this comment, we conducted an additional experiment with a similar workflow to the experiment shown in Figure 6F. Pan/99-TC virus was allowed to replicate under multi-cycle conditions from an MOI of 0.002 PFU/cell for 12 hours. After those 12 hours, another population of cells was inoculated with Pan/99-TC virus at an MOI of 0.1 PFU/cell. Both populations were then incubated under single-cycle conditions for 12 hours. The infected cells were then stained *in situ* with FlaSH reagent and the monolayer was imaged to visualize infected cells. As expected, foci of infection were observed under low MOI + multi-cycle conditions, and distributed cells were observed under high MOI + single-cycle conditions. These results are discussed within the text at Lines 249 – 253, as follows:

“First, to evaluate spatial structure under single and multi-cycle conditions, we used a reporter strain of Pan/99 virus with a tetracycline tag on the NP protein (Pan/99-NP_TC virus) and visualized infected cell monolayers. The results confirmed that single-cycle inoculation results in random dispersal of virus across cells, while multi-cycle replication proceeds in a spatially structured manner resulting in foci of infection (Fig. 6C).”

The M.STOP experiments are very clever, and the guinea pig experiments are especially intriguing. My concern here is whether the virus is really absolutely dependent upon complementation as the authors describe it. The dilution series in Figure 7C never goes low enough to demonstrate the absence of infectious units that co-express M1 and M2. It is possible that some degree of virion aggregation in the stock means that fully replication competent infectious units are always present at some level.

This is an interesting point, and we agree that some aggregation of virions may have occurred during the egg passage. We have noted this in the text at Lines 329 – 335, as follows:

“At limiting dilutions, where only 2% of cells were infected, only one sample reached the point of absolutely no co-expression between M1 and M2, suggesting that stocks may contain some aggregates of virus particles comprising fully infectious units. We cannot fully exclude this possibility, but note that co-expressing cells represented less than 10% of infected cells at this limiting dilution, suggesting that most co-expression of M1 and M2 at higher concentrations of virus is indeed mediated by co-infection.”

We also note, however, that 1) the last dilution shown in Figure 7C is the limiting dilution for this virus stock, at which ~2% of cells express any M proteins and beyond which no cells express M1 or M2; 2) an average of 90% of infected cells at this limiting dilution express only M1 or M2; and 3) there is one sample at this limiting dilution where no cells co-express M1 and M2.

The lack of absolute requirement for M2 also complicates things, as the authors point out, meaning that M2- virions may just be attenuated rather than dead in the absence of complementation. It seems possible that *in vivo* growth could entirely be driven by the M2-dead component of the M.STOP population without any complementation, making it is hard to interpret the *in vivo* results. This could be clarified by a couple of experiments. First, how does an M2-dead mutant (in which all virions express M1 and none express M2) compare with the M.STOP virus?

To address this comment, we performed transfections of 293T cells to generate a Pan/99-M1.Only virus. We were able to recover the virus, but found that it did not replicate to a detectable extent in eggs or following passage on MDCK cells. This result strongly suggests that the M1.Only subpopulation (“M2-dead component” in the reviewer’s words) of Pan/99-M.STOP virus is not the primary driver of replication *in vivo*. These results are reported in the text at Lines 336 – 341, as follows:

“M2 expression was shown previously to be non-essential for replication in cell culture^{33,34}. To evaluate the extent to which Pan/99 virus relies on M2 expression for viral multiplication in the systems used here, we generated a virus unable to express M2 using the plasmid encoding the M1.Only segment. This virus was successfully recovered and formed small plaques in MDCK cells. Multiple attempts to culture the M1.Only virus in MDCK cells and eggs failed, however, indicating the importance of M2 to viral propagation in these substrates (data not shown).”

Furthermore, we attempted to propagate this M1.Only virus in MDCK cells that stably express the M2 protein of influenza A/WSN/1933 virus (a generous gift of Dr. Andrew Pekosz), and found that the resulting virus had recombined to incorporate the RNA encoding functional WSN M2 into its genome. This recombined variant was then selected for *in vivo*. By contrast, guinea pig infection with Pan/99-M.STOP virus led to maintenance of both M1-deficient and M2-deficient sequences *in vivo* (with no additional sequences changes detected).

Second, if guinea pigs are infected with WT virus at the dose used in Figure 8A or with an 815-fold reduction to reflect the difference in infectivity seen for M.STOP in Figure 7D, would the comparative viral load data look like that in Figure 8A? In other words, to what extent are the growth curves in Figure 8A influenced by the fact that the inoculum doses for the two viruses may have differed by roughly 800-fold in terms of guinea pig ID50?

The reviewer raises an interesting point, and we performed the suggested experiment to address this question. The data revealed that the inoculum dose had little impact on the kinetics or peak titers of WT virus shed in nasal washes, only a reduction in nasal wash titers on Day 1 post-inoculation. This result suggests that the differing growth patterns observed for WT and M.STOP viruses in Figure 8A are due to their genetic differences rather than dosing differences. These results are reported in the text in Figure 8B and at Lines 374 – 380, as follows:

“Because the inoculum of WT and M.STOP viruses comprised the same dose in terms vRNA copies, but different doses in terms of GPID50, we also inoculated guinea pigs with

WT virus at doses of $8 \times \text{GPID}_{50}$ and $6.5 \times 10^3 \times \text{GPID}_{50}$ (Fig. 8B). This experiment was designed to define the contribution of the effective dose to the differences observed between M.STOP and WT viruses in guinea pigs. Similar peak titers and kinetics of shedding were observed in both groups of WT virus infected guinea pigs, indicating minimal dose dependency (Fig. 8B).”

Line 62: Should cite Russell et al (10.7554/eLife.32303) which showed heterogeneity in influenza virus gene expression here.

We have included a citation as requested, at Lines 60 – 62, as follows:

“Recent data suggest that IAV infection is not a binary state, however. Efforts to detect viral proteins and mRNAs at the single cell level have revealed significant heterogeneity in viral gene expression¹⁹⁻²².”

Line 232: Referring to cells as “semi-infected” sounds awkward. Maybe better to say the proportion of infected cells that lack the complete viral genome.

We have made the requested change, as well as removed references to “semi-infected cells” elsewhere in the text.

Reviewer #2

In Jacobs et al, the authors perform a series of elegant experimental and modeling studies to examine the complex feature of influenza viral RNA packaging efficiency, spread and reassortment. The study sheds light on many important areas of influenza biology that are hotly debated. Using genetically similar viruses, the authors examine the ‘productive presence’ (termed Pp) of individual gene segments in a single cell level. They found that the Pp was much lower than would be expected for viruses that efficiently package all eight copies of a viral RNA. Based on sophisticated modeling they calculate that only ~1.3% of a virion progeny will produce an infection where all 8 segments are replicated and present. Therefore, co-infection of cells with multiple incomplete viral genomes is necessary for replication fitness within hosts but is dependent upon the spatial diffusion of viruses within respiratory tracts and burst size. This is evident in the creative use of a novel virus that requires co-infection to replicate, the M.stop virus will only replicate when a single cell is infected with multiple viruses encoding either M1.stop or M2.stop. Inoculation of guinea pig with the M.stop viruses demonstrates that complementation of viruses with incomplete genome is capable in an experimental infection scenario, since robust viral replication in guinea pig upper respiratory tract was observed. However, these viruses are incapable of overcoming the bottleneck in transmission to recipient animals, likely because so few virions initiate a natural infection.

These studies are very intriguing and call into question the prevailing dogma that the majority of virions contain all 8 segments. Based on the data presented in this manuscript using the single cell assay, it seems there may be a continuum, where progeny virions could contain less than all 8 segments and/or upon cellular entry not all viral RNA are transported into the nucleus with equal affinity. The authors do a nice job in the discussion detailing all possible explanations for their observations and how it fits into previous work on semi-infectious particles.

We thank the reviewer for these positive comments.

1. The single cell data elucidating the modest Pp for all 8 segments (Figure 1) is very compelling. This

reviewer has three suggestions that may be useful points of discussion regarding the single cell experiments:

a. As the authors mention in the discussion, the P_p could change based on strain type. Could the authors either report the P_p for other biological strains, or would it be possible to model the reassortment potential between strains with differing P_p phenotype? Along this same line would it be possible to attribute varying P_p values to specific segments and assess whether this would alter the relationship between segments?

We agree with the reviewer that P_p values of diverse IAV strains would be of interest. Measuring P_p values is, however, labor intensive and our preliminary efforts with other strains suggest that strain-specific optimization of conditions will be needed. As the rest of the experiments are restricted to the A/Panama/2007/99 (H3N2) strain background, we believe that measuring P_p of other IAV strains is outside the scope of this work.

With respect to reassortment between strains of differing P_p , we have added a discussion of this point to the text at Lines 437 – 440 as follows:

“Interestingly, the strain-specificity of P_p is likely to influence the relative representation of genome segments when two strains of varying P_p reassort during co-infection. As lower P_p values result in successful replication of fewer segments per cell, the genomes of reassortant progeny are likely to contain more segments from the virus with higher P_p .”

The correlations shown in Fig. 1C are non-random associations between segments, i.e. the frequency with which two segments co-occur more often than would be expected by random chance. These associations between segments account for the individual P_p value of each segment and the expected frequency of co-occurrence, so absent an additional biological interaction, two segments with significantly different P_p values would be expected to co-occur less frequently due to one segment having a lower chance of delivery.

b. Does the P_p of a viral stock change over time? One would assume that a low MOI infection could produce more incomplete viral genomes since only ~1.3% would contain all 8 segments from a given single infection.

Having measured P_p in only egg-passaged Pan/99 virus stocks, we have not directly addressed this point. We would not, however, expect P_p to change over time for the following reasons: although most infected cells lack some viral gene segments under low MOI conditions, we would not expect P_p of progeny viruses to be reduced as a result. This is because all segments encode proteins essential for completion of the viral life cycle, and therefore only the rare cells harboring all eight segments would be expected to produce progeny.

c. In lieu of a helper virus, would it be feasible to perform a similar analysis on cell lines expressing the complementing polymerase complex? This would remove any bias that may be present with a co-viral infection. Presumably a similar P_p would be generated in this case as well.

We thank the reviewer for this suggestion. We agree that a complementing cell line expressing PB2, PB1, PA and NP would likely amplify genome segments of incomplete viral genomes. However, our single-cell assay relies on propagation of the virus to neighboring cells after sorting, and if the HA segment is missing from a cell, for example, then our limit of detection would be constrained by the amount of RNA present in a single infected cell. It is difficult to

predict whether the benefit of a complementing polymerase complex would outweigh the loss of RNA resulting from a lack of viral propagation. Our attempts to generate such a cell line have also been met with technical difficulty.

2. While the majority of the manuscript was very well written and clear, further explanation of some of the modeling aspects would be helpful, especially in describing figure 4 parts B and C with the diffusion constants.

We thank the reviewer for raising this point, as we hope to ensure that the modeling aspects of the paper are accessible to experimental biologists. We have expanded the description of the agent-based model in the Methods (lines 625 – 677) to be more explicit in its description of how virions emerge from infected cells, disperse to new cells, and subsequently infect them. Furthermore, we now show how the dynamics of ‘cells-infected’ and ‘virions-present’ vary with diffusion coefficient in Supplemental Figure 2.

1. Line 15 – the second ‘the’ should read then.

We thank the reviewer for catching this and have corrected the error.

2. Please state the timepoint used to generate Figure 5B.

We have included a note in the Figure 5 legend and Methods section to specify that cells were collected 12 hours post-inoculation for FACS analysis. This is reflected in the text at Lines 799 – 800, as follows:

“At 12 h post-inoculation, cells were harvested and stained for analysis of HA expression by flow cytometry.”

Reviewer #3

The authors follow-up on the prior work of their own group (Fonville et al, PLoS Path 2015, ref [24]) and others (Heldt et al, Nat Comm 2015, ref [20]; Brooke et al, JVI 2013, ref [19]) demonstrating the existence and effects of so-called ‘incomplete (influenza) viral genomes’ (IVGs) with substantial new experimental and theoretical results. The major results are:

* The development of a robust, simple, and novel experimental assay to detect the frequency at which each of the eight influenza genome segments is ultimately available for a cell's infection when that infection is accomplished with a single virion. Using this assay, the probability of occurrence of each segment is constrained to a narrow range (0.5-0.6), and a simple calculation leads to an estimate of the fraction of singly-infected cells that have a complete set (1.3 percent). [Figure 1A]

* The experimentally-measured frequency of segments are then used to parameterize a stochastic simulation (introduced in [24]), which in turn is able to fully predict the experimentally-measured frequency of reassortment in influenza virions (also from [24]). As they describe in the manuscript (line 125), this "indicates that IVGs fully account for the levels of reassortment observed."

* A relatively simple probability model for infection, parameterized by the above-measured probability of occurrence of segments, is used to predict the expected number of viral entries to yield a productive infection (3.7 entry events).

* An agent-based computational model is used to demonstrate that constricted dispersal of newly-produced virions by a cell (mediated by a finite diffusion constant in a liquid medium) can allow for high-MOI infections of nearby cells and thus allow for the regular presence of a full complement of genomic segments. This effect was found to be consistent with their in vitro experiments.

* By developing a mutant virus that required reassortment for propagation (i.e., while wild-type virions are very unlikely to contain the full complement of genomic segments, none of the constructed mutants did), they demonstrated that a relatively normal (though attenuated) infection could be established. But, they also show that transmission, which likely occurs through "low-MOI" infections by single virions, was blocked for this mutant strain, indicating that "rare complete IAV genomes may be critical for transmission to new hosts."

Each of these accomplishments is important to the field, novel, and will spur future research. The combined impact of all of them makes this a significant work. Therefore, I am very supportive of publication of this manuscript in Nature Communications.

We thank the reviewer for these positive comments.

Nevertheless, I have some suggestions for the authors:

1) In a few places, some disambiguation of the theoretical methods would be appreciated and useful to the reader. The equation used to calculate P_p (probability present of the i th segment) from the FACS results, which appears after line 624, could be explained with a few more lines. For example, "Using the relationship between MOI and the fraction of cells infected from Poisson statistics, i.e., $f = 1 - \exp(-\text{MOI})$, the probability of the i th segment can be calculated from the 96-well infections as:

$$P_{p,i} = \text{MOI}_i / \text{MOI}_{wt} = -\ln(1 - f_i) / -\ln(1 - f_{wt}) = \ln(1 - C/A) / \ln(1 - B/A)"$$

Also, it is repeatedly stated in the text that this is a "correction factor" when in fact it is just the equation from which one expects to calculate the correct value of $P_{p,i}$ (the equation is not a multiplicative "factor").

We thank the reviewer for this helpful comment. We have included the explanation suggested and included simplifications in the equation describing the calculation of P_p . Furthermore, we have removed references to this equation as a "correction factor." The revised text, from Lines 517 – 528 follows:

"Here we describe the mathematical analysis used to calculate P_p from the experimental data. This calculation is needed because, given the MOI of Pan/99-WT virus used in the experiments, an appreciable number of wells are expected to receive two or more viral genomes.

Using the relationship between MOI and the fraction of cells infected from Poisson statistics, i.e., $f = 1 - e^{-\text{MOI}}$, the probability of the i th segment being present in a singly infected cell, or $P_{p,i}$ can be calculated from the 96-well plate using the following equation:

$$P_{P,i} = \frac{\text{MOI}_i}{\text{MOI}_{wt}} = \frac{-\ln(1 - f_i)}{-\ln(1 - f_{wt})} = \frac{\ln(1 - \frac{C_i}{A})}{\ln(1 - \frac{B}{A})}$$

where A is the number of Helper⁺ cells, B is the number of WT⁺ cells (containing any WT segment), and C_i is the number of wells positive for the WT segment in question. Wells that were negative for Helper virus segments (and consequently did not contain WT segments) were excluded from analysis. For each experimental replicate, the geometric mean P_p value was calculated to represent an average P_p .”

Within the Methods section on the probability model, it would be helpful to the reader to briefly show that the basic definition of the expectation $E[v]$ (weighted average of the v values) leads to the equation after line 478, via “[$I - T$]⁽⁻¹⁾ = $I + T + T^2 + \dots$ ”.

We have updated the written description of the method to further clarify the derivation of $(I - T)^{-1}$, and added an explicit expansion $I + T + T^2 + \dots T^n$ to the equation in question. Furthermore, we have included a citation to Finite Markov Chains (Kemeny & Snell), where a more detailed derivation can be found. The text (Lines 598 – 613) follows:

“Finally, we use survival analysis to calculate the expected number of virions that must infect a cell before it receives all 8 segments. We first define T_{sub} to represent the upper-left 8x8 matrix of T (in which a cell contains 0 – 7 segments), τ_{sub} as the first 8 columns of τ_0 , and 1_{sum} as an 8x1 vector where each value is 1, which acts to sum each state into a single value. For a cell that starts with 0 segments, $\tau_{sub} = [1, 0, \dots, 0]$. The probability distribution of a cell containing 0 – 7 segments is given by $\tau_{sub} * I$ (where I is the identity matrix) for an uninfected cell, $\tau_{sub} * T_{sub}$ for 1 virion, $\tau_{sub} * T_{sub}^2$ for 2 virions, and so on. For an arbitrary number of virions (v), the distribution is given by $\tau_{sub} * T_{sub}^v$. The total probability that a cell contains 0 – 7 segments is then calculated as $\tau_{sub} * T_{sub}^v * 1_{sum}$. As more virions infect a cell, this probability converges to 0. We therefore estimate the number of virions required to fully infect a cell using the equation:

$$E(v) = \tau_{sub} * \lim_{n \rightarrow \infty} (I + T_{sub}^1 + T_{sub}^2 + \dots T_{sub}^n) * 1_{sum} = \tau_{sub} * (I - T_{sub})^{-1} * 1_{sum}$$

This summary statistic represents the number of transitions required for a cell to reach the absorbing state, or more simply, the average number of virions required to infect a cell. The variance on this quantity can be calculated as $\sigma^2 = (2 * (I - T_{sub})^{-1} - I) * \tau_{sub} - \tau_{sub, sq}$, where each element of τ_{sub} is squared to generate $\tau_{sub, sq}$. A more detailed proof of these derivations can be found in Finite Markov Chains⁴⁵.”

Finally, within the Methods the description of the "Monte Carlo simulations" should precede the probability model, since its use does in the Results, and perhaps it and the probability model should be given their own short sub-sections, as is given to the agent-based model.

We have set the “Monte Carlo simulations” section apart as requested. This section (Lines 614 – 623), which was used to generate Figure 3, discussed in Lines 174 – 188, follows the “Probabilistic model to estimate costs of incomplete genomes for cellular infectivity” section (Lines 566 – 611), which was used to generate Figure 2, discussed in Lines 147 – 172.

2) Some more details about the agent-based model would be useful, since there are some decisions that must have been made in this simulation that are not completely explained. For example, how are the rates from Table 1 implemented in the dynamics, how is the diffusion implemented (e.g., is a random

distance and a random direction vector chosen?), do the agent based dynamics match those of an ODE model (for some test cases) in the limit of high diffusion? These could be addressed in a section of the Supplemental Material, rather than expanding the Methods.

We thank the reviewer for raising this point and have added the following elements to address this deficiency:

- 1) We have provided more detail about the mechanisms of virion diffusion, infection, and attachment in the section beginning at line 625.**
- 2) We note in the Methods that the rates from Table 1 are implemented using the Poisson distribution. This explanation is reflected in the text at Lines 670 – 673, as follows:**
“At each time-step, the number of virions that bind, release, or infect, and the number of cells that die or become refractory to super-infection, are calculated using the Poisson distribution with $\lambda = \text{Rate} * N$, where Rates are described in Table 1, and N represents the number of virions or infected cells present at that time.”

Because of the attachment, detachment, death, and superinfection exclusion processes occurring in this system, generating an ODE model was not trivial. We note, however, in the Results (Lines 212 – 213) that values of $D > 10^3 \text{ um}^2/\text{s}$ provide approximately random mixing of virions, as demonstrated by the curves in Figure 4 leveling off to the right of $D = 10^3 \text{ um}^2/\text{s}$, which we have tested out to $D = 10^{12} \text{ um}^2/\text{s}$. At these $D > 10^3 \text{ um}^2/\text{s}$, the distribution of virion locations in the 100x100 grid after a diffusion event follows a uniform distribution, due to the high variance of the distribution governing particle spread and the model forcing virions that pass outside the boundaries of the grid to wrap around from the other side.

3) Uncertainties and/or confidence intervals could be associated to the values obtained from the various experiments and models and added to the text. For example the 0.58 value for P_p , the 3.7 virions value for the "infectious unit", the 11.5 PFU burst size.

We have calculated these confidence intervals and incorporated them into the main text and Figures (as noted here).

- $P_p = 0.58$ (0.57 – 0.59), Figure 1A**
- % Fully Infectious Particles = 1.22% (0 – 30.5), Figure 2B**
- Infectious Unit = 3.6 virions (1 – 6.5), Figure 2D**
- Burst Size = 11.5 PFU/cell (10.6 – 12.5), Figure 5C**

As discussed in the text, the percentage of fully infectious units was estimated as a parameter of the Bernoulli distribution, $p = P_p^8$, $\sigma^2 = p(1-p) = P_p^8(1 - P_p^8)$, assuming $P_p = 0.58$. The infectious unit was calculated by parameterizing the model “Probabilistic model to estimate costs of incomplete genomes for cellular infectivity” with $P_p = 0.58$ and calculating σ^2 as described on Lines 567 – 571, as follows:

“To define the impact of incomplete viral genomes on viral infectivity, we first considered how specific infectivity, the ratio of plaque-forming units to virus particles, changes with P_p . The proportion of virions that can form plaques, or the probability of productive infection resulting from a single virion, was estimated as $p = P_p^8$. This is a Bernoulli process with a defined probability of success or failure, so the variance was estimated as $\sigma^2 = p(1 - p) = P_p^8(1 - P_p^8)$.”

4) The Introduction provides a coherent and complete introduction to the problem at hand. This reader would have liked to hear a little bit more about the issue discussed in lines 67-71 --- i.e., the origin of IVGs, if most isolated virions are found to be complete --- either within the Introduction or in the Discussion. It is determined (and then taken for granted) throughout the manuscript that singly-infected cells have an incomplete set of viable genome segments, yet the evidence presented in those lines seems to say that virions entering cells are completed. Some speculation or discussion in the literature would be valuable here. (Although perhaps no more can be said than the statement in the Discussion on line 369.)

We have discussed this further in the text, which appears at Lines 66 – 75, as follows:

“Replication and expression of only a subset of the genome may be explained by two potential mechanisms: either the majority of particles lack one or more genome segments, or segments are readily lost in the process of infection before they can be replicated. Published data suggest that most particles contain full genomes: electron microscopy revealed eight distinct RNA segments in most virions²⁴, and FiSH-based detection of viral RNAs indicated that a virion typically contains one copy of each segment²⁵. Loss of segments following delivery of a viral genome to the target cell therefore seems likely to be an important mechanism. Inefficiencies inherent in the processes of cytoplasmic trafficking, nuclear import, and replication of incoming viral RNAs during the earliest stages of infection would all lead to loss of segments. Very likely, multiple mechanisms contribute to give rise to incomplete IAV genomes.”

5) The limited burst size of single-cycle infections is common for high-MOI ("single-cycle") infections, and is usually associated to the effects of defective interfering particles (see, e.g., Liao et al, Royal Society Interface 13: 20160412 (2016)). This should probably be commented-on here.

This is an interesting point, which we have thought about carefully. As noted at lines 696 – 698 of the text, the defective interfering (DI) particle content of virus stocks was quantified using a digital droplet PCR (ddPCR) assay and confirmed to be minimal⁴⁴. While the possibility remains that DI segments could be generated de novo during the growth assay, we note that the cited article describes a reduction in peak virus output at high MOIs, which the authors attribute to the presence of defective interfering particles. Our results, however, show that burst size saturates at high MOIs (up to 20 PFU/cell). In contrast, we would expect the presence of DI particles to result in a maximum burst size at intermediate MOI.

6) I found a few typos (e.g., lines 24 and 28 in the abstract, line 475; the variable "k" is not defined before line 468; what are the meaning of the commas/slashes in the "Values" table), so the whole text should be checked again.

We have corrected said typographical errors and scanned the manuscript again for any other errors.

Reviewers' Comments:

Reviewer #1:

Remarks to the Author:

Reviewer comments have all been effectively addressed.

Reviewer #2:

Remarks to the Author:

In the revised manuscript, Jacobs et al have added clarity to their previous study and addressed all of this Reviewer's comments. This study adds significantly to the larger conversation on the role of semi-infectious particles in influenza viral RNA packaging efficiency, spread and reassortment. The authors have used novel approaches including modeling studies to examine the complex relationship of viral gene expression and viral fitness. The study sheds light on many important areas of influenza biology that are hotly debated in a creative and unique way that will be of broad interest to the readership of Nature Communications.

Reviewer #3:

Remarks to the Author:

I remain very supportive of publication, but I have a few more comments/concerns on the manuscript. One of these issues, #1, I did not appreciate on the first reading; the remaining items concern the updated text.

=== 1 ===

The methods used for the "Monte Carlo" simulation in the "Predicted costs of incomplete genomes for population infectivity" are not clearly explained (so the results, Figure 3A and 3B) are not reproducible. Moreover, I think that there is a hidden parameter here (the number of cells in the "population") which determines the particular results presented. Finally, I think that these same results could be more clearly presented using the analytical result (or an approximation of the analytical result), which I suggest below.

If the authors plan to keep the Monte Carlo method of obtaining these results (which I do not recommend, since it seems to me the analytic method, below, is more straightforward), then the methods need to be explained a bit more. How many cells are used in a population? Is complementarity allowed (the phrase on line 180-181 "the frequency with which each CELL acquired segments was again governed by P_P" suggests no)? How exactly are the viruses "randomly distributed"? Were poisson statistics used to distribute viruses or are you just saying that the resulting distribution of viruses from a uniform random method resulted in Poisson statistics of occupation?

The Monte Carlo methods used here seem, to me, unnecessary, since the same result should be calculable analytically. If you ignore complementarity (which I guess should not be done, but since that calculation is a little easier I'll show it first), then the probability that a particular cell has a full complement of 8 segments would be:

$$p_{\text{complete}} = p_{\text{infected}} * p_{\text{segment}}^8 = [1 - \exp(-m)] * (P_P)^8$$

where P_P is the "probability present" (assumed constant across segments), m is the MOI (virus

particles delivered to the well, divided by number of cells in the population), and (Poisson) probability that a particular cell is *not* entered is $\exp(-m)$. Therefore the number of cells in the population that receive eight segments is:

$$N_{\text{complete}} = N_{\text{cells}} * p_{\text{complete}} = N_{\text{cells}} * [1 - \exp(-m)] * (P_P)^8$$

where N_{cells} is the number of cells in a population. If N_{complete} is greater than one, this represents a successful infection of the cell population. If N_{complete} is less than one, then this fraction is the "percent of populations in which at least one cell contains 8 segments". Thus, the functions in Figure 3A could be plotted as:

$$F(\text{MOI}) = \min[N_{\text{complete}}, 1]$$

with a separate plot for every P_P value and those of 3B could be found from the intersections of $F(\text{MOI})$ with 0.5.

Thus, although the probability of infection (and the graph in Figure 6A) does not depend on the number of cells, the quantities plotted in Figures 3A and 3B definitely do. In other words, if you are calculating the probability that a population will have one cell infected, then that probability will increase as the number of available cells increases (as long as the MOI is held fixed). It should therefore be noted that these results correspond to one particular chosen value of N_{cells} , or a different quantity should be plotted that does not depend on N_{cells} .

To account for complementarity, one should be able to use the Poisson probability of n virion entries, i.e.,

$$p(n) = (m^n / n!) \exp(-m)$$

to write the probability that a cell receives the complete complement as

$$p_{\text{complete}} = \lim(N \rightarrow \infty) \text{Sum}[n=1 \text{ to } N] \{ p(n) * p_{\text{eight}}(n) \}$$

where $p_{\text{eight}}(n)$ is the probability of getting eight viable segments in n entry events, i.e.,

$$p_{\text{eight}}(n) = [1 - (1 - P_P)^n]$$

This expression (using some large value of N) could be used to calculate the functions plotted in 3A and 3B without resorting to Monte Carlo methods. Again, the results will depend on the choice of N_{cells} .

One last point about this section. The functions plotted in 3A were fitted with logistic curves to obtain (one of?) the graphs in 3B. But given the "min" function above, this does not seem to be appropriate. Unless I am mistaken, the graphs in 3A should not be smooth curves, but piecewise continuous curves given by the min function. If an analytical method is used, then there is no need to fit Monte Carlo data.

=== 2 ===

I think the introduction of confidence intervals on many quantities is a useful addition. For some of them, however, it is not clear how they were obtained. Finding the variance does not necessarily imply a 95% confidence interval, but only the variance calculations are mentioned in the Methods.

Also, the 95% confidence interval given for P_P seems to be a very misleading descriptor of the data shown in Figure 1A. I would suggest that the mean values of each P_P (currently shown with their confidence intervals in Figure 1A) be quoted in the figure caption. Significances could be stated for their comparisons. Then, in the text, one could calculate the mean P_P value and the standard error in the mean. I do not understand how the [0.57-0.59] interval was obtained, and it does not seem to be a very useful quantity.

=== 3 ===

It seems that for most of the manuscript (except for Figure 1), the authors use a single value of P_P (the average value 0.58). At the point that this value is calculated, it should be mentioned why they will be using a single average value henceforth. Do the authors assume that the P_P values are independent of segment? If so, this should be stated. Also, the variable P_P is used both for the probability of one particular segment being present and for any one of the segments. Perhaps P_{Pi} should be used initially and then P_P defined as the average value and the assumed constant probability for any segment.

=== 4 ===

A few smaller points:

* Line 168, it is not clear that the relationship is truly "inversely proportional", is it? Similarly, on line 185, the phrase "increases exponentially as P_P decreases" seems imprecise.

* Should it be mentioned somewhere what is meant by a PFU? I assume this is found by standard dilutions on MDCK cells, but given the central findings of this paper, a value of 1 PFU would actually imply a much higher concentration of IVGs, correct?

* The use of "wells" and "cells" is confusing on line 518. Should it actually be "Helper+ wells"?

* Reference to the Finite Markov Chains book on line 604 seems non-standard and the citation [44] is incomplete.

* The colored points on the graphs in Figures 2B and 2D are confusing because it looks like the line is a fit to data points. Is it possible to represent these values in another way?

Reviewer #1

Reviewer comments have all been effectively addressed.

We are glad the reviewer found our changes sufficient to address their concerns.

Reviewer #2

In the revised manuscript, Jacobs et al have added clarity to their previous study and addressed all of this Reviewer's comments. This study adds significantly to the larger conversation on the role of semi-infectious particles in influenza viral RNA packaging efficiency, spread and reassortment. The authors have used novel approaches including modeling studies to examine the complex relationship of viral gene expression and viral fitness. The study sheds light on many important areas of influenza biology that are hotly debated in a creative and unique way that will be of broad interest to the readership of Nature Communications.

We thank this reviewer for their supportive comments and contextualization of this work's contribution to the literature.

Reviewer #3

I remain very supportive of publication, but I have a few more comments/concerns on the manuscript. One of these issues, #1, I did not appreciate on the first reading; the remaining items concern the updated text.

=== 1 ===

The methods used for the "Monte Carlo" simulation in the "Predicted costs of incomplete genomes for population infectivity" are not clearly explained (so the results, Figure 3A and 3B) are not reproducible. Moreover, I think that there is a hidden parameter here (the number of cells in the "population") which determines the particular results presented. Finally, I think that these same results could be more clearly presented using the analytical result (or an approximation of the analytical result), which I suggest below.

If the authors plan to keep the Monte Carlo method of obtaining these results (which I do not recommend, since it seems to me the analytic method, below, is more straightforward), then the methods need to be explained a bit more. How many cells are used in a population? Is complementarity allowed (the phrase on line 180-181 "the frequency with which each CELL acquired segments was again governed by P_P" suggests no)? How exactly are the viruses "randomly distributed"? Were poisson statistics used to distribute viruses or are you just saying that the resulting distribution of viruses from a uniform random method resulted in Poisson statistics of occupation?

The Monte Carlo methods used here seem, to me, unnecessary, since the same result should be calculable analytically. If you ignore complementarity (which I guess should not be done, but

since that calculation is a little easier I'll show it first), then the probability that a particular cell has a full complement of 8 segments would be:

$$p_{\text{complete}} = p_{\text{infected}} * p_{\text{segment}}^8 = [1 - \exp(-m)] * (P_P)^8$$

where P_P is the "probability present" (assumed constant across segments), m is the MOI (virus particles delivered to the well, divided by number of cells in the population), and (Poisson) probability that a particular cell is *not* entered is $\exp(-m)$. Therefore the number of cells in the population that receive eight segments is:

$$N_{\text{complete}} = N_{\text{cells}} * p_{\text{complete}} = N_{\text{cells}} * [1 - \exp(-m)] * (P_P)^8$$

where N_{cells} is the number of cells in a population. If N_{complete} is greater than one, this represents a successful infection of the cell population. If N_{complete} is less than one, then this fraction is the "percent of populations in which at least one cell contains 8 segments". Thus, the functions in Figure 3A could be plotted as:

$$F(\text{MOI}) = \min[N_{\text{complete}}, 1]$$

with a separate plot for every P_P value and those of 3B could be found from the intersections of $F(\text{MOI})$ with 0.5.

Thus, although the probability of infection (and the graph in Figure 6A) does not depend on the number of cells, the quantities plotted in Figures 3A and 3B definitely do. In other words, if you are calculating the probability that a population will have one cell infected, then that probability will increase as the number of available cells increases (as long as the MOI is held fixed). It should therefore be noted that these results correspond to one particular chosen value of N_{cells} , or a different quantity should be plotted that does not depend on N_{cells} .

To account for complementarity, one should be able to use the Poisson probability of n virion entries, i.e.,

$$p(n) = (m^n / n!) \exp(-m)$$

to write the probability that a cell receives the complete complement as

$$p_{\text{complete}} = \lim(N \rightarrow \infty) \text{Sum}[n=1 \text{ to } N] \{ p(n) * p_{\text{eight}}(n) \}$$

where $p_{\text{eight}}(n)$ is the probability of getting eight viable segments in n entry events, i.e.,

$$p_{\text{eight}}(n) = [1 - (1 - P_P)^n]$$

This expression (using some large value of N) could be used to calculate the functions plotted in 3A and 3B without resorting to Monte Carlo methods. Again, the results will depend on the choice of N_{cells} .

We thank the reviewer for this insightful comment, which was helpful in improving both the accuracy and accessibility of Figure 3. We have remade Figures 3A, 3B, and 6A using the analytical solution suggested, and have clarified the new method in the text at lines 619 – 639 as follows:

Probabilistic model to estimate costs of IVGs for population infectivity

To define the impact of incomplete viral genomes on the ability of a virus population to establish infection in a population of cells, the probabilistic model described above was adapted to account for the Poisson distribution of virions among a population of cells. It is assumed that one productively infected cell produces enough virions to infect other cells in subsequent rounds of replication, and so establishing infection in a population of cells requires that at least one cell receive all 8 genome segments. For a given MOI, the probability of a cell being infected by v virions follows the Poisson distribution $p(v) = \frac{MOI^v e^{-MOI}}{v!}$. At each v , the probability that a cell received any given segment is equal to $1 - (1 - P_p)^v$, and so the probability that a cell is productively infected after infection with v virions is $p(8|v) = (1 - (1 - P_p)^v)^8$. The sum of the joint probabilities $p(v) * p(8|v)$ across all values of v gives the probability that any given cell is productively infected: $\lim_{N \rightarrow \infty} \sum_{v=1}^N p(v) * p(8|v)$. Multiplying this probability by the number of cells in the population gives the expected number of cells infected, and the probability of the population becoming infected is equal to this value or 1, whichever is lower. The ID_{50} was estimated as the lowest MOI yielding a probability $\geq 50\%$. A similar analysis was used to estimate the ID_{50} when complementation was not allowed, with the $p(8|v)$ function being modified to $p(8|v) = 1 - (1 - P_p^8)^v$ to reflect the fact that only complete viral genomes could initiate infection. Finally, the percentage of infected cells that contained incomplete viral genomes was calculated by estimating the probability that a cell infected by v virions contained between 1 and 7 segments, $p(1-7|v) = 1 - (1 - (1 - P_p)^v)^8$, and determining the total proportion of infected cells containing IVGs using the equation $\% \text{ Cells with IVGs} = \lim_{N \rightarrow \infty} \sum_{v=1}^N p(v) * \frac{p(1-7|v)}{p(1-7|v) + p(8|v)} * 100$.

Furthermore, we have verified that the population ID_{50} depends on the number of cells, and noted this in the text at lines 268 – 270 as follows:

“It is important to note that these calculations depend on the number of cells in the population being considered, and so the probability of establishing infection may be influenced by the surface area of target tissue.”

One last point about this section. The functions plotted in 3A were fitted with logistic curves to obtain (one of?) the graphs in 3B. But given the "min" function above, this does not seem to be appropriate. Unless I am mistaken, the graphs in 3A should not be smooth curves, but piecewise continuous curves given by the min function. If an analytical method is used, then there is no need to fit Monte Carlo data.

The reviewer is correct in concluding that the graphs in Figure 3A should not be smooth logistic curves. In previous versions of the manuscript, logistic regression was used to estimate the point at which each line intersected with 50%. The lines in Figure 3A are now piecewise continuous curves, as predicted, and the ID_{50} values shown in Figure 3B are the lowest MOI that yields an infection probability $\geq 50\%$ for each value of P_P . In calculating infection probabilities in 3A, \log_{10} MOI was varied from -6 to 1 by increments of 0.01 , so these estimates are accurate to 3 significant figures.

==== 2 ====

I think the introduction of confidence intervals on many quantities is a useful addition. For some of them, however, it is not clear how they were obtained. Finding the variance does not necessarily imply a 95% confidence interval, but only the variance calculations are mentioned in the Methods.

We have updated the Figure legends where appropriate to indicate how 95% confidence intervals were determined. Specifically, 95% confidence intervals are $\text{mean} \pm 1.96 * \text{S.D.}$ for theoretical estimates (Figs 2B, 2D), $\text{mean} \pm 1.96 * \text{S.E.}$ (Fig 1A), or 95% confidence intervals of regression lines (Figs 4A, 4B, 4C, 4D, 6B, 6D, 7C, Supp. 6C).

Also, the 95% confidence interval given for P_P seems to be a very misleading descriptor of the data shown in Figure 1A. I would suggest that the mean values of each P_P (currently shown with their confidence intervals in Figure 1A) be quoted in the figure caption. Significances could be stated for their comparisons. Then, in the text, one could calculate the mean P_P value and the standard error in the mean. I do not understand how the $[0.57-0.59]$ interval was obtained, and it does not seem to be a very useful quantity.

We now include individual P_P values for each segment in Figure 1A, as well as reported an average (geometric mean) P_P value with 95% confidence interval ($\text{mean} \pm 1.96 * \text{S.E.}$), of 0.58 ($0.54 - 0.61$) based on 13 experimentally measured P_P values. We comment on this in the text at lines 200 – 203 as follows:

“ ...an average P_P value was estimated for each experimental replicate by calculating the geometric mean of the eight segment-specific $P_{P,i}$ values. The arithmetic mean of each of these 13 summary P_P values was 0.58 (95% C.I. $0.54 - 0.61$).”

==== 3 ====

It seems that for most of the manuscript (except for Figure 1), the authors use a single value of P_P (the average value 0.58). At the point that this value is calculated, it should be mentioned why they will be using a single average value henceforth. Do the authors assume that the P_P values are independent of segment? If so, this should be stated. Also, the variable P_P is used both for the probability of one particular segment being present and for any one of the segments.

Perhaps P_{Pi} should be used initially and then P_P defined as the average value and the assumed constant probability for any segment.

We have adopted the P_{Pi} notation as suggested by the reviewer and agree that this distinction is helpful. We have also explained the rationale for use of a single value at the point of its introduction and given a more detailed explanation of how this value was calculated. Essentially, the single value of P_P was used for simplicity. This clarification is reflected in the text at lines 156 – 160 and 199 – 204, as follows:

“We termed the resultant parameter “Probability Present”, and refer to it hereafter as P_P . Segment-specific values are referred to as $P_{P,i}$ where $i = 1 - 8$, while P_P refers to the average P_P value across all segments, which is calculated as the geometric mean of eight segment-specific values to reflect the fact that productive infection requires independent delivery of all eight genome segments.” Lines 156 – 160

“Given the independence of vRNP delivery and the similarity between $P_{P,i}$ values, we calculated an average P_P value for use in subsequent analyses. Specifically, an average P_P value was estimated for each experimental replicate by calculating the geometric mean of the eight segment-specific $P_{P,i}$ values. The arithmetic mean of each of these 13 summary P_P values was 0.58 (95% C.I. 0.54 – 0.61). The models described below use the average P_P value of 0.58 for simplicity.” Lines 199 – 204

=== 4 ===

A few smaller points:

* Line 168, it is not clear that the relationship is truly "inversely proportional", is it? Similarly, on line 185, the phrase "increases exponentially as P_P decreases" seems imprecise.

We have updated the text to remove these imprecisions at Lines 227 – 228 and 242 – 261, respectively, as follows:

“We see that the number of virions comprising an infectious unit increases sharply at low values of P_P .” Lines 227 – 228

“Indeed, when we estimated the MOI required for a virus of a given P_P to have a 50% chance of infecting a population, we observed that the ID_{50} increases on a logarithmic scale as P_P decreases (Fig. 3B).” Lines 242 – 261

* Should it be mentioned somewhere what is meant by a PFU? I assume this is found by standard dilutions on MDCK cells, but given the central findings of this paper, a value of 1 PFU would actually imply a much higher concentration of IVGs, correct?

The reviewer’s assumption of the meaning of PFU is correct, along with the corollary that 1 PFU implies the presence of more IVGs. We have clarified this in the text at lines 532 – 541, as follows:

“If these 1.22% of virions comprise the plaque-forming units present in a virus population, then the total number of virions present in a population is equal to 82 times the number of plaque-forming units (because $1 / 0.0122 = 82$).”

* The use of "wells" and "cells" is confusing on line 518. Should it actually be "Helper+ wells"?

We have changed the text to reflect “Helper+ wells” and “WT+ wells.” Furthermore, we have clarified that each “well” of the 96-well plate represents the viral RNA from the single cell that was sorted into it at lines 626 – 629, as follows:

“The presence or absence of different viral genome segments was measured by qRT-PCR, with each well of a 96-well plate representing the viral RNA that was present in the cell that was initially sorted into the plate.”

* Reference to the Finite Markov Chains book on line 604 seems non-standard and the citation [44] is incomplete.

We have corrected the citation, following the format provided on *Nature Communications* website:

“Kemeny, J.G.; Snell, J.L. *Finite Markov Chains*. (Springer-Verlag, New York, 1976).”

* The colored points on the graphs in Figures 2B and 2D are confusing because it looks like the line is a fit to data points. Is it possible to represent these values in another way?

We have moved the points representing experimentally measured P_P values to the bottom of the plot, with colored lines connecting the points to their corresponding place on the theoretical line.

Reviewers' Comments:

Reviewer #3:

Remarks to the Author:

The authors have addressed all of my comments well, and I'm glad that the adjustments could be made so quickly. I enthusiastically support publication.

I just have a few small comments, but I do not need to see any further versions.

* I'm still not sure about some of the 95% confidence interval statements. The expression mean +/- 1.96*sigma would give the confidence interval for a normal distribution, but not necessarily the distributions seen here. Perhaps only the sigma should be given?

* There is a typo in the equation at line 638.

* The denominator in the sum at lines 642-643 is just one, correct?

* The sentence in lines 448-449 is now slightly ambiguous because of the insertion of the "82 times" in the previous sentence. Perhaps reword.

Reviewer #3

The authors have addressed all of my comments well, and I'm glad that the adjustments could be made so quickly. I enthusiastically support publication.

We thank the reviewer for these positive comments and support of publication.

I just have a few small comments, but I do not need to see any further versions.

* I'm still not sure about some of the 95% confidence interval statements. The expression mean $\pm 1.96 \times \sigma$ would give the confidence interval for a normal distribution, but not necessarily the distributions seen here. Perhaps only the sigma should be given?

We have adjusted the listed confidence intervals to (mean \pm SD).

* There is a typo in the equation at line 638.

We have corrected the error and clarified the equation at Line 620 in the text, as follows:

$$"p(1-7 | v) = 1 - p(8 | v) - p(0 | v) = 1 - (1 - P_p^8)^v - (1 - P_p)^{8v}"$$

* The denominator in the sum at lines 642-643 is just one, correct?

The denominator is equal to $1 - p(0 | v)$, which represents the probability that a cell contains at least one genome segment following infection. There is some probability, especially at low values of v and P_p , that one or more virions infect a cell but no segments are successfully delivered. This probability is captured in the term $p(0 | v)$.

* The sentence in lines 448-449 is now slightly ambiguous because of the insertion of the "82 times" in the previous sentence. Perhaps reword.

We see the reviewer's point and have clarified that the estimate of 1.22% of virions being lower than other published results.